# Inference-Time Scaling for Flow Models via Stochastic Generation and Rollover Budget Forcing

**Jaihoon Kim**[*]    **Taehoon Yoon**[*]    **Jisung Hwang**[*]    **Minhyuk Sung**

KAIST

{jh27kim,taehoon,4011hjs,mhsung}@kaist.ac.kr

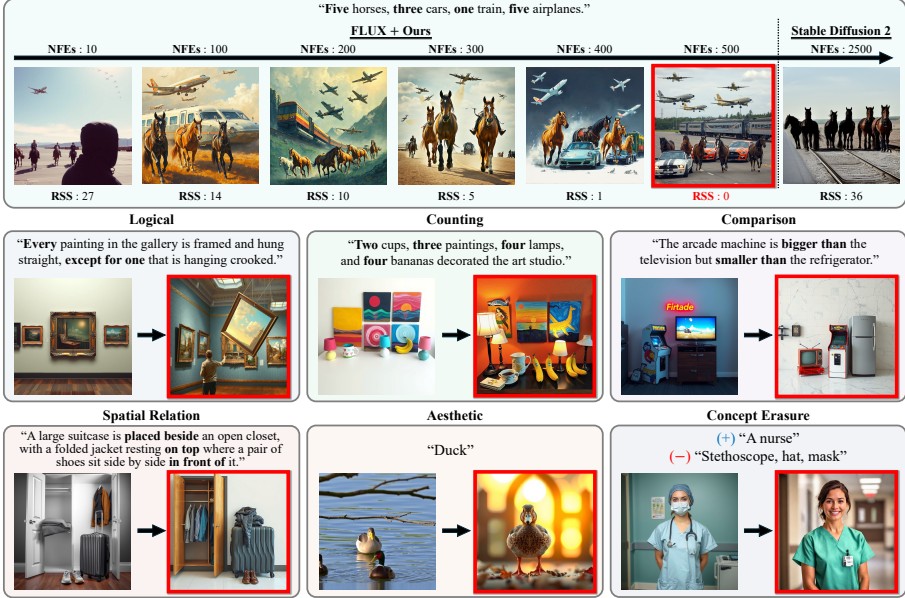

Figure 1: **Diverse applications of our inference-time scaling method.** Pretrained flow models struggle to generate images that align with complex prompts (left side of each case), whereas our inference-time scaling effectively extends their capabilities to achieve precise alignment (red box).

## Abstract

We propose an inference-time scaling approach for pretrained flow models. Recently, inference-time scaling has gained significant attention in LLMs and diffusion models, improving sample quality or better aligning outputs with user preferences by leveraging additional computation. For diffusion models, particle sampling has allowed more efficient scaling due to the stochasticity at intermediate denoising steps. On the contrary, while flow models have gained popularity–offering faster generation and high-quality outputs–efficient inference-time scaling methods used for diffusion models cannot be directly applied due to their deterministic generative process. To enable efficient inference-time scaling for flow models, we propose three key ideas: 1) SDE-based generation, enabling particle sampling in flow models, 2) Interpolant conversion, broadening the search space, and 3) Rollover Budget Forcing (RBF), maximizing compute utilization. Our experiments show that SDE-based generation and variance-preserving (VP) interpolant-based generation, improves the performance of particle sampling methods for inference-time scaling in flow models. Additionally, we demonstrate that RBF with VP-SDE achieves the best performance, outperforming all previous inference-time scaling approaches. Project page: flow-inference-time-scaling.github.io.

[*]Equal contribution.

39th Conference on Neural Information Processing Systems (NeurIPS 2025).

# 1 Introduction

Over the past years, scaling laws of AI models have mainly focused on increasing model size and training data. However, recent advancements have shifted attention toward *inference-time scaling* [57, 59], leveraging computational resources during inference to enhance model performance. OpenAI o1 [42] and DeepSeek R1 [11] exemplify this approach, demonstrating consistent output improvements with increased inference computation. Recent research in LLMs [41] attempting to replicate such improvements has introduced *test-time budget forcing*, achieving high efficiency with limited token sampling during inference.

For diffusion models [54, 56], which are widely used for generation tasks, research on inference-time scaling has been growing in the context of reward-based sampling [23, 30, 53]. Given a reward function that measures alignment with user preferences [26] or output quality [50, 31], the goal is to find the sample from the learned data distribution that best aligns with the reward through repeated sampling. Fig. 1 showcases diverse applications of inference-time scaling using our method, enabling the generation of faithful images that accurately align with complex user descriptions involving objects quantities, logical relationships, and conceptual attributes. Notably, naïve generation from text-to-image models [48, 28] often fails to fully meet user specifications, highlighting the effectiveness of inference-time scaling.

Our goal in this work is to extend the inference-time scaling capabilities of diffusion models to flow models. Flow models [32] power state-of-the-art image [15, 28] and video generation [7, 72], achieving high-quality synthesis with few inference steps, enabled by trajectory stratification techniques during training [35]. Beyond just speed, recent pretrained flow models, equipped with enhanced text-image embeddings [46] and advanced architectures [15], significantly outperform previous pretrained diffusion models in both image and video generation quality.

Despite their advantages in generating high-quality results more efficiently than diffusion models, flow models have an inherent limitation in the context of inference-time scaling. Due to their ODE-based deterministic generative process, they cannot directly incorporate particle sampling at intermediate steps, a key mechanism for effective inference-time scaling in diffusion models. Building on the formulation of stochastic interpolant framework [1], we adopt an SDE-based sampling method for flow models at inference-time, enabling particle sampling for reward alignment.

To further expand the exploration space, we consider not only stochasticity but also the choice of the *interpolant*. While typical flow models use a linear interpolant, diffusion models commonly adopt a Variance-Preserving (VP) interpolant [56, 18]. Inspired by this, for the first time, we incorporate the VP interpolant into the particle sampling of flow models and demonstrate its effectiveness in increasing sample diversity, enhancing the likelihood of discovering high-reward samples.

We emphasize that while we propose converting the generative process of a pretrained flow model to align with that of diffusion models—i.e.,VP-SDE-based generation—inference-time scaling with flow models offers significant advantages over diffusion models. Flow models, particularly those with rectification fine-tuning [35, 36], produce much clearer expected outputs at intermediate steps, enabling more precise future reward estimation and, in turn, more effective particle sampling.

We additionally explore a strategy for tight budget enforcement in terms of the number of function evaluations (NFEs) of the velocity prediction network. Previous particle-sampling-based inference-time scaling approaches for diffusion models [30, 53] allocate the NFEs budget *uniformly* across timesteps in the generative process, which we empirically found to be ineffective in practice. To optimize budget utilization, we propose *Rollover Budget Forcing*, a method that adaptively reallocates NFEs across timesteps. Specifically, we perform a denoising step upon identifying a new particle with a higher expected future reward and allocate the remaining NFEs to subsequent timesteps.

Experimentally, we demonstrate that our inference-time SDE conversion and VP interpolant conversion enable efficient particle sampling in flow models, leading to consistent improvements in reward alignment across two challenging tasks: compositional text-to-image generation and quantity-aware image generation. Additionally, our Rollover Budget Forcing (RBF) provides further performance gains, outperforming all previous particle sampling approaches. We also demonstrate that for differentiable rewards, such as aesthetic image generation, integrating RBF with a gradient-based method [8] creates a synergistic effect, leading to further performance improvements.

In summary, we introduce an inference-time scaling for flow models, analyzing three key factors:

- ODE vs. SDE: We introduce an *SDE generative process* for flow models to enable particle sampling.

- Interpolant: We demonstrate that replacing the linear interpolant of flow models with *Variance Preserving interpolant* expands the search space, facilitating the discovery of higher-reward samples.

- NFEs Allocation: We propose *Rollover Budget Forcing* that adaptively allocates NFEs across timesteps to ensure efficient utilization of the available compute budget.

## 2 Related Work

### 2.1 Reward Alignment in Diffusion Models

In the literature of diffusion models, reward alignment approaches can be broadly categorized into fine-tuning-based methods [5, 69, 62, 9, 43, 67] and inference-time-scaling-based methods [30, 53, 13, 64, 6]. While fine-tuning diffusion models enables the generation of samples aligned with user preferences, it requires fine-tuning for each task, potentially limiting scalability. In contrast, inference-time scaling approaches offer a significant advantage as they can be applied to any reward without requiring additional fine-tuning. Moreover, inference-time scaling can also be applied to fine-tuned models to further enhance alignment with the reward. Since our proposed approach is an inference-time scaling method, we focus our review on related literature in this domain.

When the reward is differentiable, gradient-based methods [8, 3, 71, 16, 17, 63, 4] have been extensively studied. We note that inference-time scaling can be integrated with gradient-based approaches to achieve synergistic performance improvements.

### 2.2 Particle Sampling with Diffusion Models

The simplest iterative sampling method that can be applied to any generative model is Best-of-N (BoN) [57, 59, 58], which generates $N$ samples and selects the one with the highest reward. For diffusion models, however, incorporating particle sampling during the denoising process has been shown to be far more effective than naïve BoN [53, 30]. This idea has been further developed through various approaches that sample particles at intermediate steps. For instance, SVDD [30] proposed selecting the particle with the highest reward at every step. CoDe [53] extends this idea by selecting the highest-reward particle only at specific intervals. On the other hand, methods based on Sequential Monte Carlo (SMC) [64, 6, 23, 13] employ a probabilistic selection approach, in which particles are sampled from a multinomial distribution according to their importance weights. Despite the success of particle sampling approaches for diffusion models, they have not been applicable to flow models due to the absence of stochasticity in their generative process. In this work, we present the first inference-time scaling method for flow models based on particle sampling by introducing stochasticity into the generative process and further increasing sampling diversity through trajectory modification.

### 2.3 Inference-Time Scaling with Flow Models

To our knowledge, Search over Paths (SoP) [39] is the only inference-time scaling method proposed for flow models, which applies a forward kernel to sample particles from the deterministic sampling process of flow models. However, SoP does not explore the possibility of modifying the reverse kernel, which could enable the application of more diverse particle-sampling-based methods [30, 53, 23]. To the best of our knowledge, we are the *first* to investigate the application of particle sampling to flow models through the lens of the reverse kernel.

## 3 Problem Definition and Background

### 3.1 Inference-Time Reward Alignment

Given a pretrained flow model that maps the source distribution, a standard Gaussian distribution $p_1$, into the data distribution $p_0$, our objective is to generate high-reward samples $\mathbf{x}_0 \in \mathbb{R}^d$ from the pretrained flow model without additional training–a task known as inference-time reward alignment. We denote the given reward function as $r : \mathbb{R}^d \to \mathbb{R}$, which measures text alignment or user preference for a generated sample. Following previous works [27, 60, 61], our objective can be formulated as finding the following target distribution:

$$p_0^* = \arg\max_q \; \mathbb{E}_{\mathbf{x}_0 \sim q} \underbrace{[r(\mathbf{x}_0)]}_{\text{Reward}} - \beta \underbrace{\mathcal{D}_{\text{KL}}[q\|p_0]}_{\text{KL Regularization}}, \tag{1}$$

which maximizes the expected reward while the KL divergence term prevents $p_0^*(\mathbf{x}_0)$ from deviating too far from $p_0(\mathbf{x}_0)$, with its strength controlled by the hyperparameter $\beta$. As shown in previous work [45], the target distribution $p_0^*$ can be computed as:

$$p_0^*(\mathbf{x}_0) = \frac{1}{Z} p_0(\mathbf{x}_0) \exp\left(\frac{r(\mathbf{x}_0)}{\beta}\right), \tag{2}$$

where $Z$ is a normalization constant. We present details in Appendix A.1. However, sampling from the target distribution is non-trivial.

A notable approach for sampling from the target distribution is *particle sampling*, which maintains a set of candidate samples—referred to as particles—and iteratively propagates high-reward samples while discarding lower-reward ones. When combined with the denoising process of diffusion models, particle sampling can improve the efficiency of limited computational resources in inference-time scaling. In the next section, we review particle sampling methods used in diffusion models and, *for the first time,* we explore insights for adapting them to flow models.

### 3.2 Particle Sampling Using Diffusion Model

A pretrained diffusion model generates data by drawing an initial sample from the standard Gaussian distribution and iteratively sampling from the learned conditional distribution $p_\theta(\mathbf{x}_{t-\Delta t}|\mathbf{x}_t)$. Building on this, previous works [29, 61] have shown that data from the target distribution in Eq. 2 can be generated by performing the same denoising process while replacing the conditional distribution $p_\theta(\mathbf{x}_{t-\Delta t}|\mathbf{x}_t)$ with the *optimal policy*:

$$p_\theta^*(\mathbf{x}_{t-\Delta t}|\mathbf{x}_t) = \frac{p_\theta(\mathbf{x}_{t-\Delta t}|\mathbf{x}_t) \exp\left(\frac{v(\mathbf{x}_{t-\Delta t})}{\beta}\right)}{\int p_\theta(\mathbf{x}_{t-\Delta t}|\mathbf{x}_t) \exp\left(\frac{v(\mathbf{x}_{t-\Delta t})}{\beta}\right) \mathrm{d}\mathbf{x}_{t-\Delta t}}, \tag{3}$$

where the details are presented in Appendix A.2. We denote $v(\cdot) : \mathbb{R}^d \to \mathbb{R}$ as the optimal value function that estimates the expected future reward of the generated samples at current timestep. Following previous works [8, 23, 30, 3], we approximate the value function using the posterior mean computed via Tweedie's formula [47], given by $v(\mathbf{x}_t) \approx r(\mathbf{x}_{0|t})$, where $\mathbf{x}_{0|t} := \mathbb{E}_{\mathbf{x}_0 \sim p_\theta(\mathbf{x}_0|\mathbf{x}_t)}[\mathbf{x}_0]$.

Since directly sampling from the optimal policy distribution in Eq. 3 is nontrivial, one can first approximate the distribution using importance sampling while taking $p_\theta(\mathbf{x}_{t-\Delta t}|\mathbf{x}_t)$ as the *proposal distribution*:

$$p_\theta^*(\mathbf{x}_{t-\Delta t}|\mathbf{x}_t) \approx \sum_{i=1}^{K} \frac{w_{t-\Delta t}^{(i)}}{\sum_{j=1}^{K} w_{t-\Delta t}^{(j)}} \delta_{\mathbf{x}_{t-\Delta t}^{(i)}}, \quad \{\mathbf{x}_{t-\Delta t}^{(i)}\}_{i=1}^{K} \sim p_\theta(\mathbf{x}_{t-\Delta t}|\mathbf{x}_t), \tag{4}$$

where $K$ is the number of particles, $w_{t-\Delta t}^{(i)} = \exp\left(v(\mathbf{x}_{t-\Delta t}^{(i)})/\beta\right)$ is the weight, and $\delta_{\mathbf{x}_{t-\Delta t}^{(i)}}$ is a Dirac distribution. SVDD [30] proposed an approximate sampling method for the optimal policy by selecting the sample with the largest weight from Eq. 4.

Notably, a key factor in seeking high-reward samples using particle sampling is defining the proposal distribution to sufficiently cover the distribution of high-reward samples. Consider a scenario where high-reward samples reside in a low density region of the original data distribution, which is common when generating complex or highly specific samples that deviate from the mode of the pretrained model distribution. In this case, the proposal distribution must have a sufficiently large variance to effectively explore these low density regions. This highlights the importance of the *stochasticity* of the proposal distribution, which has been instrumental in the successful adoption of particle sampling in diffusion models. In contrast, flow models [32] employ a *deterministic* sampling process, where all particles $\mathbf{x}_{t-\Delta t}$ drawn from $\mathbf{x}_t$ are identical. This restricts the applicability of particle sampling methods in flow models. One of the main contributions is the investigation of how these particle sampling methods can be efficiently applied to flow models.

To this end, we propose an inference-time approach that introduces stochasticity into the generative process of flow models to enable particle sampling. We first transform the deterministic sampling process of flow models into a stochastic process (Sec. 4.2). We further identify a sampling trajectory that expands the search space of the flow models (Sec. 4.3). Note that while stochastic sampling and trajectory conversion have been studied in prior works, their primary goals have been to improve sample quality [68, 70, 49, 24, 38] or to accelerate inference [19, 52, 51, 22]. To the best of our

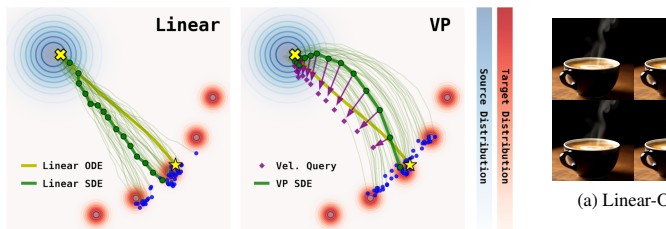

Figure 2: **Comparison of Linear-ODE, Linear-SDE, and VP-SDE.** The visualization shows how trajectories evolve under different dynamics starting from the same noise latent.

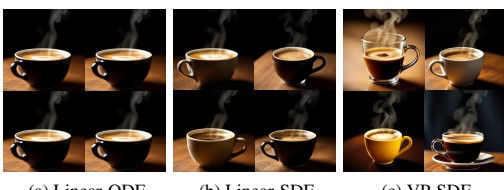

(a) Linear-ODE     (b) Linear-SDE     (c) VP-SDE

Figure 3: **Sample diversity test using FLUX [28] under linear and VP interpolant.** All samples share the same initial latent. Prompt: *"A steaming cup of coffee"*.

knowledge, we are the first to investigate sampling stochasticity and trajectory conversion for efficient particle-based sampling in flow models.

Additionally, previous particle sampling methods in diffusion models allocated a fixed computational budget (i.e., a uniform number of particles) across all denoising timesteps, potentially limiting exploration. We explore sampling with the rollover strategy, which adaptively allocates compute across timesteps during the sampling process (Sec. 6).

## 4 SDE-Based Particle Sampling in Flow Models

In this section, we review flow and diffusion models within the unified stochastic interpolant framework (Sec. 4.1) and introduce our inference-time approaches for efficient particle sampling in flow models (Sec. 4.2 and 4.3).

### 4.1 Background: Stochastic Interpolant Framework

At the core of both diffusion and flow models is the construction of probability paths $\{p_t\}_{0 \le t \le 1}$, where $\mathbf{x}_t \sim p_t$ serves as a bridge between $\mathbf{x}_1 \sim p_1$ and $\mathbf{x}_0 \sim p_0$:

$$\mathbf{x}_t = \alpha_t \mathbf{x}_0 + \sigma_t \mathbf{x}_1, \tag{5}$$

where $\alpha_t$ and $\sigma_t$ are smooth functions satisfying $\alpha_0 = \sigma_1 = 1$, $\alpha_1 = \sigma_0 = 0$, and $\dot{\alpha}_t < 0, \dot{\sigma}_t > 0$; we denote the dot as a time derivative. This formulation provides a flexible choice of *interpolant* $(\alpha_t, \sigma_t)$ which determines the sampling trajectory.

### 4.2 Inference-Time SDE Conversion

Flow models [32, 35] learn the velocity field $u_t : \mathbb{R}^d \to \mathbb{R}^d$, which enables sampling of $\mathbf{x}_0$ by solving the Probability Flow-ODE [56] backward in time:

$$\mathrm{d}\mathbf{x}_t = u_t(\mathbf{x}_t)\mathrm{d}t. \tag{6}$$

The deterministic process in Eq. 6 accelerates the sampling process enabling few-step generation of high-fidelity samples. However, as discussed in Sec. 3.2, the deterministic nature of this sampling process limits the applicability of particle sampling in flow models.

To address this, we transform the deterministic sampling process into a stochastic process. The reverse-time SDE that shares the same marginal densities as the deterministic process in Eq. 6:

$$\mathrm{d}\mathbf{x}_t = \mathbf{f}_t(\mathbf{x}_t)\mathrm{d}t + g_t\mathrm{d}\mathbf{w}, \quad \mathbf{f}_t(\mathbf{x}_t) = u_t(\mathbf{x}_t) - \frac{g_t^2}{2}\nabla \log p_t(\mathbf{x}_t), \tag{7}$$

where $\mathbf{f}_t(\mathbf{x}_t)$ and $g_t$ represent the drift and diffusion coefficient, respectively, and $\mathbf{w}$ is the standard Wiener process. This conversion introduces a noise schedule $g_t$, which can be freely chosen. Although SiT [38] arrives at the same conclusion, we provide a more comprehensive proof in Appendix B. In our case, we set $g_t = t^2$, scaled by a factor of 3. Note that in the case where $g_t = 0$ the process reduces to deterministic sampling in Eq. 6.

Using the velocity $u_t(\mathbf{x}_t)$ predicted by a pretrained flow model, the score function $\nabla \log p_t(\mathbf{x}_t)$ appearing in the drift coefficient $\mathbf{f}_t(\mathbf{x}_t)$ can be computed as:

$$\nabla \log p_t(\mathbf{x}_t) = \frac{1}{\sigma_t} \frac{\alpha_t u_t(\mathbf{x}_t) - \dot{\alpha}_t \mathbf{x}_t}{\dot{\alpha}_t \sigma_t - \alpha_t \dot{\sigma}_t}. \tag{8}$$

This enables the conversion of the deterministic sampling to stochastic sampling, which we refer to as inference-time SDE conversion. Given the drift coefficient term $\mathbf{f}_t(\mathbf{x}_t)$ and diffusion coefficient $g_t$, the proposal distribution in the discrete-time domain is derived as follows:

$$p_\theta(\mathbf{x}_{t-\Delta t}|\mathbf{x}_t) = \mathcal{N}(\mathbf{x}_t - \mathbf{f}_t(\mathbf{x}_t)\Delta t, \; g_t^2 \Delta t\, \mathbf{I}). \tag{9}$$

While previous works have proposed converting an SDE to an ODE to improve sampling efficiency [22, 55, 37, 56], the reverse approach—transforming an ODE into an SDE—has received relatively less attention and has primarily focused on improving sample quality [68, 38]. To the best of our knowledge, this work is the *first* to explore SDE conversion in flow models specifically to expand the search space of proposal distribution for efficient particle sampling.

Since flow models utilize the linear interpolant ($\alpha_t = 1 - t, \sigma_t = t$), we refer to the generative processes of the flow models using Eq. 6 and Eq. 7 as Linear-ODE and Linear-SDE, respectively. In Fig. 2 (left), we visualize the sampling trajectories of Linear-ODE and Linear-SDE. The samples generated using Linear-ODE are identical and collapse to a single point, restricting exploration. In contrast, Linear-SDE introduces sample variance, allowing for broader exploration and increasing the likelihood of discovering high-reward samples.

In Fig. 3 (a-b), we visualize images sampled from Linear-ODE and Linear-SDE using FLUX [28]. As discussed previously, the particles drawn from the proposal distribution of Linear-ODE are identical. In contrast, Linear-SDE introduces variation across different particles, thereby expanding the search space for identifying high-reward samples. In the next section, we introduce *inference-time interpolant conversion*, which further increases the search space.

### 4.3 Inference-Time Interpolant Conversion

To further expand the search space of Linear-SDE, we draw inspiration from the effective use of particle sampling in diffusion models, where we identified a key difference: the *interpolant*. While the forward process in diffusion models follows the Variance Preserving (VP) interpolant ($\alpha_t = \exp^{-\frac{1}{2}\int_0^t \beta_s \mathrm{d}s}, \sigma_t = \sqrt{1 - \exp^{-\int_0^t \beta_s \mathrm{d}s}}$), with $\beta_s$ denoting a predefined variance schedule, flow models adopt a linear interpolant.

As shown in the previous works [33, 52], we note that given a velocity model $u_t$ based on an interpolant ($\alpha_t, \sigma_t$) (e.g., linear), one can transform the vector field and generate a sample based on a new interpolant ($\bar{\alpha}_s, \bar{\sigma}_s$) (e.g., VP) at inference-time. The two paths $\bar{\mathbf{x}}_s = \bar{\alpha}_s \mathbf{x}_0 + \bar{\sigma}_s \mathbf{x}_1$ and $\mathbf{x}_t = \alpha_t \mathbf{x}_0 + \sigma_t \mathbf{x}_1$ are connected through scale-time transformation:

$$\bar{\mathbf{x}}_s = c_s \mathbf{x}_{t_s} \quad t_s = \rho^{-1}(\bar{\rho}(s)) \quad c_s = \bar{\sigma}_s/\sigma_{t_s}, \tag{10}$$

where $\rho(t) = \frac{\alpha_t}{\sigma_t}$ and $\bar{\rho}(s) = \frac{\bar{\alpha}_s}{\bar{\sigma}_s}$ define the signal-to-noise ratio of the original and the new interpolant, respectively. The velocity for the new interpolant is given as:

$$\bar{u}_s(\bar{\mathbf{x}}_s) = \frac{\dot{c}_s}{c_s}\bar{\mathbf{x}}_s + c_s \dot{t}_s u_{t_s}\left(\frac{\bar{\mathbf{x}}_s}{c_s}\right), \quad \dot{c}_s = \frac{\sigma_{t_s}\dot{\bar{\sigma}}_s - \bar{\sigma}_s\dot{\sigma}_{t_s}\dot{t}_s}{\sigma_{t_s}^2} \quad \dot{t}_s = \frac{\sigma_{t_s}^2\left(\bar{\sigma}_s\dot{\bar{\alpha}}_s - \bar{\alpha}_s\dot{\bar{\sigma}}_s\right)}{\bar{\sigma}_s^2\left(\sigma_{t_s}\dot{\alpha}_{t_s} - \alpha_{t_s}\dot{\sigma}_{t_s}\right)}. \tag{11}$$

Plugging the transformed velocity into the proposal distribution in Eq. 9 after computing the score using Eq. 8 gives our efficient proposal distribution.

$$\bar{p}_\theta(\bar{\mathbf{x}}_{s-\Delta s}|\bar{\mathbf{x}}_s) = \mathcal{N}\left(\bar{\mathbf{x}}_s - \left[\bar{u}_s(\bar{\mathbf{x}}_s) - \frac{g_s^2}{2}\nabla \log \bar{p}_s(\bar{\mathbf{x}}_s)\right]\Delta s, \; g_s^2\Delta s\, \mathbf{I}\right). \tag{12}$$

Since the new trajectory follows the VP interpolant, we refer to this as VP-SDE. We visualize VP-SDE sampling in Fig. 2 (right). At inference-time, we query the velocity of the new interpolant from the original interpolant (purple arrow). In Fig. 3 (c), we visualize the sample diversity under VP-SDE using FLUX [28] which generates more diverse samples than Linear-SDE. This property of VP-SDE effectively expands the search space, improving particle sampling efficiency in flow models. In Sec. 5, we provide further analysis on how interpolant conversion contributes to sample diversity.

Previous works focused on interpolant conversion that enables stable training [49, 12, 24] and accelerated inference [51, 22, 52]. We utilize interpolant conversion to enhance the sample diversity in particle sampling, which has *not* been unexplored before. Importantly, while we modify the generative process of flow models to align with that of diffusion models, inference-time scaling with flow models still provides distinct advantages. The rectified trajectories of flow models [35, 36, 28] allow for a much clearer posterior mean, leading to more precise future reward estimation and, in turn, more effective particle filtering.

# 5 Analysis of the Interpolant Conversion and Sample Diversity

In this section, we analyze how the interpolant conversion affects the variance of the proposal distribution and explain why VP-SDE yields higher sample diversity than Linear-SDE, as illustrated in Fig. 3. To investigate this behavior, Fig. 4 visualizes the log-SNR ($\log(\alpha_t^2/\sigma_t^2)$) of commonly used interpolants including linear and VP across timesteps $t \in (0, 1)$.

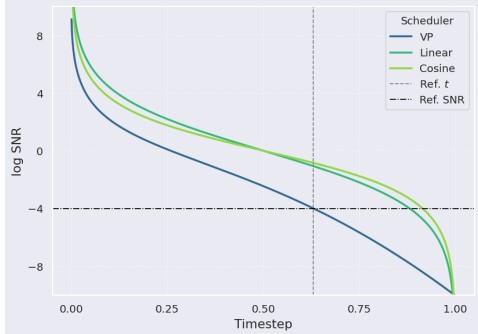

Then consider initializing the Linear-SDE timesteps $\{t_s\}_{0 \le s \le 1}$ using the timestep conversion in Eq. 10. This ensures that the log-SNR of the corresponding latents $\mathbf{x}_{t_s}$ matches that of the VP-SDE latents $\bar{\mathbf{x}}_s$ at each step (see the horizontal dashed line in Fig. 4). Under this condition, the proposal distributions of the two processes are expressed as follows:

Figure 4: **Interpolant log-SNR.** Dashed lines show a reference SNR and timestep.

$$\text{Linear-SDE } (t_s): \quad p_\theta(\mathbf{x}_{t_s - \Delta t_s} \mid \mathbf{x}_{t_s}) = \mathcal{N}\left(\mathbf{x}_{t_s} - \mathbf{f}_{t_s}(\mathbf{x}_{t_s})\Delta t_s, \ g_{t_s}^2 \Delta t_s \mathbf{I}\right)$$

$$\text{VP-SDE}: \quad \bar{p}_\theta(\bar{\mathbf{x}}_{s-\Delta s} \mid \bar{\mathbf{x}}_s) = \mathcal{N}\left(\bar{\mathbf{x}}_s - \bar{\mathbf{f}}_s(\bar{\mathbf{x}}_s)\Delta s, \ g_s^2 \Delta s \mathbf{I}\right) = \bar{p}_\theta(\cdot \mid c_s \mathbf{x}_{t_s}).$$

Since $\Delta t_s < \Delta s$ at early denoising steps, timestep conversion results in smaller variance $g_{t_s}^2 \Delta t_s$ than VP-SDE for a fixed diffusion coefficient. Hence, to match the variance of Linear-SDE ($t_s$) to that of VP-SDE, one can scale the diffusion coefficient to $g_{t_s}' = g_s/c_s\sqrt{\Delta s/\Delta t_s}$. Note that this scaling significantly increases the stochasticity at early denoising steps as $c_s \approx 1$ and $\sqrt{\Delta s/\Delta t_s} \gg 1$. While this scaling can enhance sample diversity, applying it in isolation injects excessive noise, causing samples to deviate from the predefined denoising trajectory and ultimately degrading output quality.

In fact, interpolant conversion counteracts excessive noise injection by pairing diffusion coefficient scaling with timestep conversion. The two mechanisms act synergistically to increase the sample diversity without harming the sample quality. In Sec. 7, we validate this analysis with an ablation study that isolates the effect of each factor.

**Comparison under Identical Timestep and Diffusion Coefficient.** We next analyze the case where both Linear-SDE and VP-SDE operate under identical, fixed timestep schedules and diffusion coefficients. While this setting yields identical proposal distribution variances, Fig. 4 shows that the VP interpolant maintains a consistently lower log-SNR, indicating that at any given timestep, VP-SDE samples contain a larger noise component (see the vertical dashed line). Consequently, the VP-SDE proposal distribution effectively samples from a noisier latent at each step, resulting in higher sample diversity. This reflects the observation that noisier latents produce more diverse samples [40]. While this work focuses on the interpolant perspective, a systematic exploration of timestep scheduling and diffusion coefficient scaling remains a promising direction for future research.

# 6 Rollover Budget Forcing

In the previous sections, we have introduced our inference-time approaches to expand the search space of proposal distribution. Here, we propose a new budget-forcing strategy to maximize the use of limited compute in inference-time scaling. To the best of our knowledge, previous particle sampling methods for diffusion models [30, 53] employ a fixed number of particles across all denoising steps. However, our analysis shows that this uniform allocation may lead to inefficiency, where the NFEs required at each denoising step to obtain a sample $\mathbf{x}_{t-\Delta t}$ with a higher reward than the current sample $\mathbf{x}_t$ significantly varies across different runs. We present the analysis results in Appendix C.

This motivates us to adopt a *rollover* strategy that adaptively allocates NFEs across timesteps. Given a total NFEs budget, the NFEs quota $Q$ is allocated uniformly across timesteps. Then at each timestep, if a particle $\mathbf{x}_{t-\Delta t}$ yields a higher reward than the current sample $\mathbf{x}_t$ within the quota, we immediately proceed to the next timestep from the newly identified high-reward sample, rolling over the remaining NFEs to the next step. If the allocated quota is exhausted without identifying a better sample, we select the particle with the highest expected future reward from the current set, following the strategy used in SVDD [30]. The pseudocode of RBF is presented in Appendix D. In the next section, we demonstrate the effectiveness of RBF, along with SDE conversion and interpolant conversion.

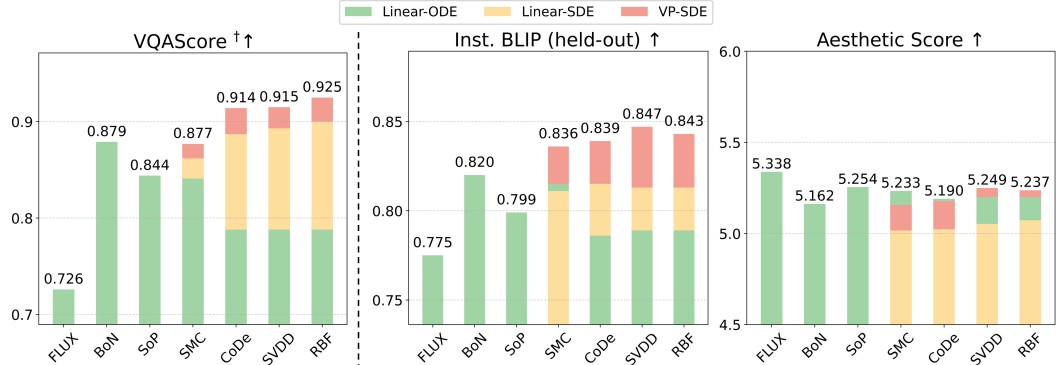

Figure 5: **Quantitative results of compositional text-to-image generation.** [†] denotes the given reward used in inference-time scaling (left). Notably, performance consistently improves from Linear-ODE to Linear-SDE and VP-SDE for both given and held-out rewards (left, middle), without significant quality degradation, as evidenced by the comparable aesthetic score [50] (right).

# 7 Applications

In this section, we present the experimental results of particle sampling methods for inference-time reward alignment. In Appendix, we present i) implementation details of the search algorithms, ii) aesthetic image generation, iii) comparisons between diffusion and flow models, iv) scaling behavior comparison of Best-of-N (BoN) and RBF, and v) additional qualitative results.

## 7.1 Experiment Setup

**Tasks.** In this section, we present the results for the following applications: compositional text-to-image generation and quantity-aware image generation, where the rewards are non-differentiable. For the differentiable reward case, we consider aesthetic image generation (Appendix E.1). In compositional text-to-image generation, we use all 121 text prompts from GenAI-Bench [21] that contain three or more advanced compositional elements. For quantity-aware image generation, we use 100 randomly sampled prompts from T2I-CompBench++ [20] numeracy category.

For all applications, we use FLUX [28] as the pretrained flow model. We fix the total number of function evaluations (NFEs) to 500 and set the number of denoising steps to 10, which allocates 50 NFEs per denoising step. As a reference, we also include the results of the base pretrained models without inference-time scaling. Additionally, we present a comparison between flow models and diffusion models in Appendix E.2.

**Baselines.** We evaluate inference-time search algorithms discussed in Sec. 2, including Best-of-N (BoN), Search over Paths (SoP) [39], SMC [23], CoDe [53], and SVDD [30]. We categorize BoN and SoP as Linear-ODE-based methods, as their generative processes follow the deterministic process in Eq. 6. For SMC, we adopt DAS [23]; however, when the reward is non-differentiable, we use the reverse transition kernel of the pretrained model as the proposal distribution.

## 7.2 Compositional Text-to-Image Generation

**Evaluation Metrics.** In this work, we refer to the reward used for inference-time scaling as the given reward. Here, the given reward is VQAScore, measured with CLIP-FlanT5 [31], which evaluates text-image alignment. For the held-out reward, which is not used during inference, we evaluate the score using a different model, InstructBLIP [10]. Additionally, we evaluate aesthetic score [50] to assess the quality of the generated images.

**Inference-Time SDE and Interpolant Conversion.** The quantitative and qualitative results of compositional text-to-image generation are presented in Fig. 5 and Fig. 6, respectively. As discussed in Sec. 4.2, the deterministic sampling process in flow models limits the effectiveness of particle sampling, whereas introducing stochasticity significantly expands the search space and improves performance—highlighting a key contribution of our work: enabling effective particle sampling in flow models. The results in Fig. 5 support this finding, showing that Linear-SDE (yellow) consistently improves the given reward (left in Fig. 5) over the Linear-ODE (green) across all particle sampling methods, even surpassing BoN and SoP [39], which were previously the only available inference-time

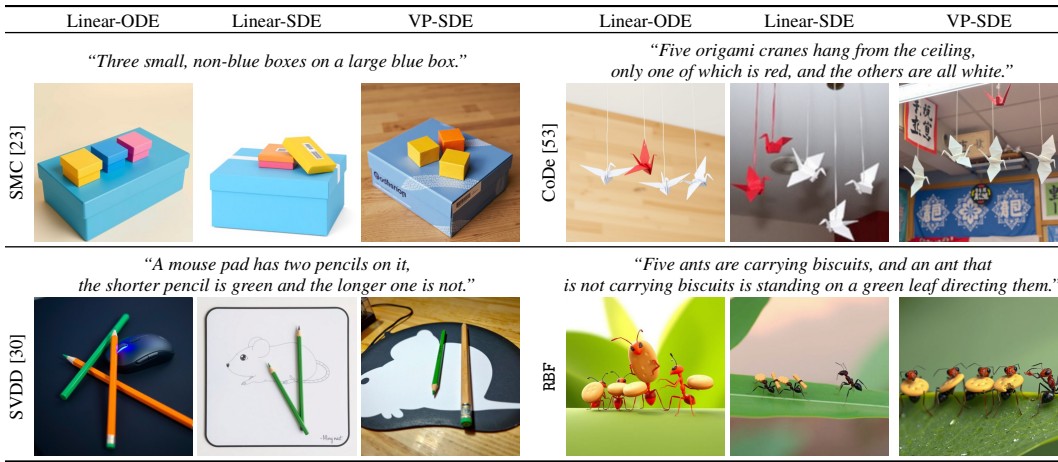

| Linear-ODE | Linear-SDE | VP-SDE | Linear-ODE | Linear-SDE | VP-SDE |

Figure 6: **Qualitative results of compositional text-to-image generation.** We use VQAScore [31], which measures text-image alignment, as the given reward for inference-time scaling. SDE and interpolant conversion enable more effective exploration during inference, enhancing the performance of all particle sampling methods [23, 53, 30], including RBF.

scaling approaches for ODE-based flow models. Additionally, through inference-time interpolant conversion, VP-SDE (red) further improves performance across all particle sampling methods on both given and held-out rewards (left, middle in Fig. 5) by expanding the search space, demonstrating the effectiveness of our proposed distribution. Notably, particle sampling methods with Linear-SDE and VP-SDE generate high-reward samples without significantly compromising image quality, as evidenced by aesthetic scores that remain comparable to the base FLUX model [28] (right in Fig. 5). Qualitatively, SDE conversion and interpolant conversion shown in Fig. 6 bring consistent performance improvements (see Appendix G.1 for additional results).

**Rollover Budget Forcing.** As discussed in Sec. 6, instead of fixing the number of particles throughout the denoising process, we explore adaptive budget allocation through RBF. We demonstrate that budget forcing provides additional performance improvements, outperforming the previous particle sampling methods in the given reward (left in Fig. 5). We present qualitative comparisons of inference-time scaling methods in Appendix G.2.

**Ablation study of interpolant conversion.** Building on the analysis in Sec. 5, we examine how interpolant conversion contributes to sample diversity and reward alignment through its two underlying mechanisms, timestep conversion and diffusion coefficient scaling. Tab. 1 extends the results of Fig. 5 by isolating the effect of each component to sample diversity, measured by LPIPS-MPD [23], and reward alignment.

Table 1: **Ablation Study of Interpolant Conversion.** [†] denotes the given reward.

| Method | LPIPS-MPD ↑ | VQAScore[†] ↑ | Inst. BLIP ↑ |
|---|---|---|---|
| Linear-ODE | – | 0.788 | 0.789 |
| Linear-SDE | 0.158 | 0.900 | 0.813 |
| + Adapt. Time. | 0.270 | 0.908 | 0.813 |
| + Adapt. Diff. | 0.429 | 0.702 | 0.571 |
| VP-SDE | **0.509** | **0.925** | **0.843** |

We observe that timestep conversion (row 3) yields only modest diversity gains: the benefit of sampling at lower log-SNR (Fig. 4) is offset by smaller discretization steps that reduce proposal variance, limiting improvements in reward alignment. On the other hand, applying diffusion coefficient scaling without timestep conversion (row 4) increases sample diversity but simultaneously leads to a significant drop in reward alignment indicating excessive noise injection. Lastly, the VP-SDE interpolant conversion (row 5) synergistically combines both components, achieving high sample diversity without sacrificing quality and consequently yielding the highest reward.

### 7.3 Quantity-Aware Image Generation

**Evaluation Metrics.** Here, the given reward is the negation of the Residual Sum of Squares (RSS) between the target counts and the detected object counts, computed using GroundingDINO [34] and SAM [25] (details in Appendix F). Additionally, we report object count accuracy, which evaluates whether all object quantities are correctly shown in the image. For the held-out reward, we report

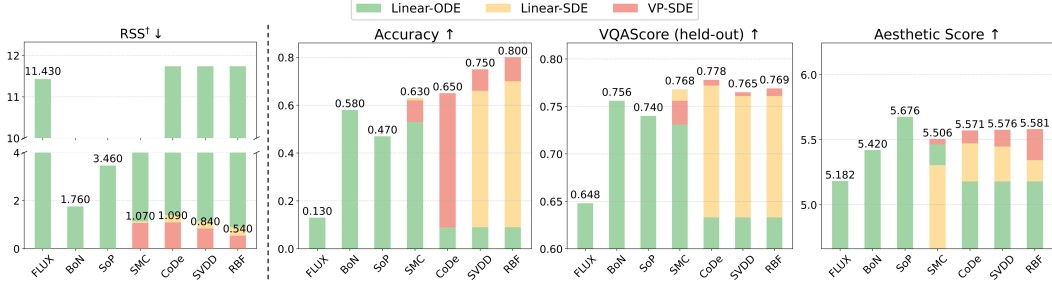

Figure 7: **Quantitative results of quantity-aware image generation.** $\dagger$ denotes the given reward, RSS [34], with the y-axis truncated for better visualization (left). We observe consistent performance improvements by converting Linear-ODE to Linear-SDE, and VP-SDE for most cases.

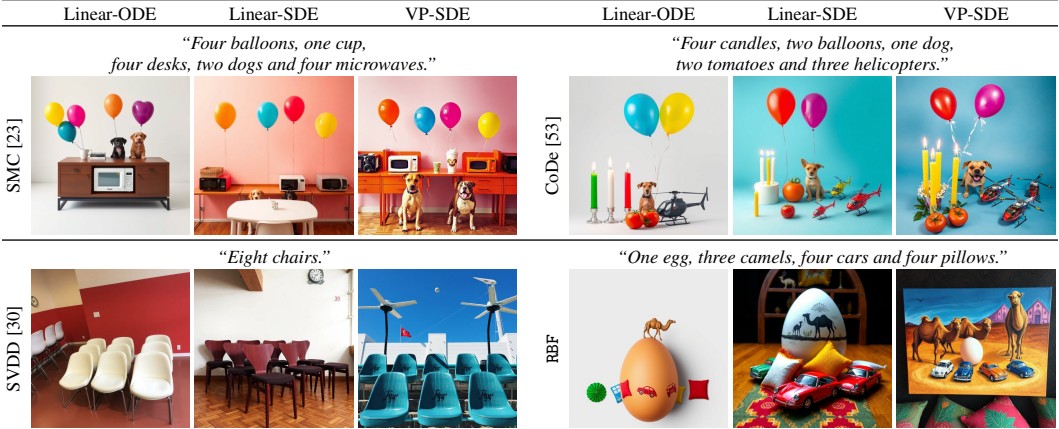

Figure 8: **Qualitative results of quantity-aware image generation.** At inference-time, we guide generation using the negation of RSS [34] (Residual Sum of Squares) as the given reward, which measures the discrepancy between detected and target object counts. SDE and interpolant conversion expands the search space to identify high reward samples.

VQAScore measured with CLIP-FlanT5 [31]. As in the previous application, we evaluate the quality of the generated images using the aesthetic score [50].

**Results.** The quantitative and qualitative results are presented in Fig. 7 and Fig. 8, respectively. The trend in Fig. 7 align with those in Sec. 7.2, demonstrating that SDE conversion and interpolant conversion synergistically enhance the identification of high-reward samples. Notably, particle sampling methods with Linear-SDE already outperform Linear-ODE-based methods (BoN and SoP [39]), while interpolant conversion further improves accuracy, achieving a $4 \sim 6\times$ improvement over the base model [28]. Our RBF achieves the highest accuracy, outperforming all other particle sampling methods. Qualitatively, Fig. 8 shows that SDE and interpolant conversion effectively identify high-reward samples that accurately match the specified object categories and quantities. Additional qualitative comparisons of the inference-time scaling methods are provided in Appendix G.2.

## 8 Conclusion and Limitation

We introduced a novel inference-time scaling method for flow models with three key contributions: (1) ODE-to-SDE conversion for particle sampling in flow models, (2) Linear-to-VP interpolant conversion for enhanced diversity and search efficiency, and (3) Rollover Budget Forcing (RBF) for adaptive compute allocation. We demonstrated the effectiveness of VP-SDE-based generation in applying off-the-shelf particle sampling to flow models and showed that our RBF combined with VP-SDE generation outperforms previous methods. However, our method introduces additional inference-time overhead, which could become a bottleneck when the base model prediction is computationally intensive. Also, since the pretrained model may have been trained on uncurated datasets, our approach may produce undesirable outputs upon malicious attempts.

## Acknowledgments

This work was supported by the NRF of Korea (RS-2023-00209723); IITP grants (RS-2022-II220594, RS-2023-00227592, RS-2024-00399817, RS-2025-25441313, RS-2025-25443318, RS-2025-02653113); and the Technology Innovation Program (RS-2025-02317326), all funded by the Korean government (MSIT and MOTIE), as well as by the DRB-KAIST SketchTheFuture Research Center.

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

# Appendix

## A  Proofs

### A.1  Derivation of the Target Distribution

From Eq. 1, we obtain the target distribution $p_0^*$, which maximizes the reward while maintaining proximity to the distribution of the pretrained model $p_0$:

$$p_0^*(\mathbf{x}_0) = \arg\max_q \ \mathbb{E}_{\mathbf{x}_0 \sim q} \left[ r(\mathbf{x}_0) \right] - \beta \mathcal{D}_{\mathrm{KL}} \left[ q \| p_0 \right],$$

$$= \arg\max_q \mathbb{E}_{\mathbf{x}_0 \sim q} \left[ r(\mathbf{x}_0) - \beta \log \frac{q(\mathbf{x}_0)}{p_0(\mathbf{x}_0)} \right]$$

$$= \arg\min_q \mathbb{E}_{\mathbf{x}_0 \sim q} \left[ \log \frac{q(\mathbf{x}_0)}{p_0(\mathbf{x}_0)} - \frac{1}{\beta} r(\mathbf{x}_0) \right]$$

$$= \arg\min_q \int q(\mathbf{x}_0) \log \frac{q(\mathbf{x}_0)}{p_0(\mathbf{x}_0)} \mathrm{d}\mathbf{x}_0 - \frac{1}{\beta} \int q(\mathbf{x}_0) r(\mathbf{x}_0) \mathrm{d}\mathbf{x}_0.$$

This can be solved via calculus of variation where the functional $\mathcal{J}$ is given as follows:

$$\mathcal{J} \left[ q(\mathbf{x}_0) \right] := \int q(\mathbf{x}_0) \left( \log \frac{q(\mathbf{x}_0)}{p_0(\mathbf{x}_0)} - \frac{1}{\beta} r(\mathbf{x}_0) \right) \mathrm{d}\mathbf{x}_0.$$

Substituting $\tilde{q}(\mathbf{x}_0, \epsilon) := q(\mathbf{x}_0) + \epsilon \eta(\mathbf{x}_0)$ gives:

$$\mathcal{J} \left[ \tilde{q}(\mathbf{x}_0, \epsilon) \right] = \int \tilde{q}(\mathbf{x}_0, \epsilon) \left( \log \frac{\tilde{q}(\mathbf{x}_0, \epsilon)}{p_0(\mathbf{x}_0)} - \frac{1}{\beta} r(\mathbf{x}_0) \right) \mathrm{d}\mathbf{x}_0,$$

where $\eta(\mathbf{x}_0)$ is an arbitrary smooth function, and $\epsilon$ is a scalar parameter.

Introducing a Lagrange multiplier $\lambda$ to constraint $\int q(\mathbf{x}_0) \mathrm{d}\mathbf{x}_0 = 1$ gives:

$$\mathcal{J} \left[ \tilde{q}(\mathbf{x}_0, \epsilon) \right] = \int \tilde{q}(\mathbf{x}_0, \epsilon) \left( \log \frac{\tilde{q}(\mathbf{x}_0, \epsilon)}{p_0(\mathbf{x}_0)} - \frac{1}{\beta} r(\mathbf{x}_0) \right) + \lambda \tilde{q}(\mathbf{x}_0, \epsilon) \mathrm{d}\mathbf{x}_0$$

$$:= \int f\{\tilde{q}; \mathbf{x}_0\} \mathrm{d}\mathbf{x}_0$$

Then the problem boils down to finding a function $\tilde{q}(\mathbf{x}_0, \epsilon)$ satisfying:

$$\left. \frac{\partial \mathcal{J}}{\partial \epsilon} \right|_{\epsilon=0} = 0$$

This can be solved using the Euler-Lagrange equation:

$$\frac{\partial f}{\partial q} - \frac{\mathrm{d}}{\mathrm{d}\mathbf{x}_0} \frac{\partial f}{\partial q'} = 0,$$

where $q'$ is a derivative of $q$ with respect to $\mathbf{x}_0$ and tilde notation is dropped since the condition is to be satisfied at $\epsilon = 0$.

Note that $q'$ does not appear in $f$, so the Euler-Lagrange equation simplifies to:

$$\frac{\partial f}{\partial q} = \frac{\partial}{\partial q} \left( q(\mathbf{x}_0) \left( \log \frac{q(\mathbf{x}_0)}{p_0(\mathbf{x}_0)} - \frac{1}{\beta} r(\mathbf{x}_0) \right) + \lambda q(\mathbf{x}_0) \right) = 0$$

$$= \log \frac{q(\mathbf{x}_0)}{p_0(\mathbf{x}_0)} - \frac{1}{\beta} r(\mathbf{x}_0) + 1 + \lambda = 0. \tag{13}$$

Solving Eq. 13 gives the target distribution $p_0^*$, which minimizes the objective function in Eq. 1:

$$p_0^*(\mathbf{x}_0) = p_0(\mathbf{x}_0) \exp\left( \frac{r(\mathbf{x}_0)}{\beta} - 1 - \lambda \right) \tag{14}$$

Lastly, the Lagrangian multiplier $\lambda$ is obtained from the normalization constraint, $\exp(\lambda) = \int p_0(\mathbf{x}_0) \exp\left(\frac{r(\mathbf{x}_0)}{\beta} - 1\right) \mathrm{d}\mathbf{x}_0$. Plugging this into Eq. 14 gives the target distribution presented in Eq. 2:

$$p_0^*(\mathbf{x}_0) = \frac{p_0(\mathbf{x}_0) \exp\left(\frac{r(\mathbf{x}_0)}{\beta}\right)}{\int p_0(\mathbf{x}_0) \exp\left(\frac{r(\mathbf{x}_0)}{\beta}\right) \mathrm{d}\mathbf{x}_0}, \tag{15}$$

### A.2 Derivation of the Optimal Policy

Here, we provide the derivations of the optimal policy given in Eq. 3 for completeness, which is proposed in previous works [60, 61].

To sample from the target distribution defined in Eq. 15, previous studies utilize an optimal policy $p_\theta^*(\mathbf{x}_{t-\Delta t}|\mathbf{x}_t)$. The optimal value function $v(\mathbf{x}_t)$ is defined as the expected future reward at current timestep $t$:

$$v(\mathbf{x}_t) = \beta \log \mathbb{E}_{\mathbf{x}_0 \sim p_\theta(\mathbf{x}_0|\mathbf{x}_t)} \left[\exp\left(\frac{r(\mathbf{x}_0)}{\beta}\right)\right] \tag{16}$$

The optimal policy is the policy that maximizes the objective function:

$$p_\theta^*(\mathbf{x}_{t-\Delta t}|\mathbf{x}_t) = \underset{q(\cdot|\mathbf{x}_t)}{\arg\max} \, \mathbb{E}_{\mathbf{x}_{t-\Delta t} \sim q(\cdot|\mathbf{x}_t)} \left[v(\mathbf{x}_{t-\Delta t})\right] - \beta \mathcal{D}_{\mathrm{KL}} \left[q(\cdot|\mathbf{x}_t)\|p_\theta(\cdot|\mathbf{x}_t)\right]$$

$$= \frac{p_\theta(\mathbf{x}_{t-\Delta t}|\mathbf{x}_t) \exp\left(\frac{1}{\beta}v(\mathbf{x}_{t-\Delta t})\right)}{\int p_\theta(\mathbf{x}_{t-\Delta t}|\mathbf{x}_t) \exp\left(\frac{1}{\beta}v(\mathbf{x}_{t-\Delta t})\right) d\mathbf{x}_{t-\Delta t}} \tag{17}$$

$$= \frac{p_\theta(\mathbf{x}_{t-\Delta t}|\mathbf{x}_t) \exp\left(\frac{1}{\beta}v(\mathbf{x}_{t-\Delta t})\right)}{\exp\left(\frac{1}{\beta}v(\mathbf{x}_t)\right)} \tag{18}$$

where the last equality follows from the soft-Bellman equations [61]. For completeness, we present the theorem.

**Theorem 1.** (Theorem 1 of Uehara *et al.* [61]). *The induced distribution of the optimal policy in Eq. 17 is the target distribution in Eq. 15.*

$$p_0^*(\mathbf{x}_0) = \int \left\{ p_1(\mathbf{x}_1) \prod_{s=T}^{1} p_\theta^*(\mathbf{x}_{\frac{s}{T}-\frac{1}{T}}|\mathbf{x}_{\frac{s}{T}}) \right\} d\mathbf{x}_{\frac{1}{T}:1}.$$

However, computing the optimal value function in Eq. 16 is non-trivial. Hence, we follow the previous works [23, 30] and approximate it using the posterior mean $\mathbf{x}_{0|t} := \mathbb{E}_{\mathbf{x}_0 \sim p_\theta(\mathbf{x}_0|\mathbf{x}_t)}[\mathbf{x}_0]$:

$$v(\mathbf{x}_t) = \beta \log \left(\int \exp\left(\frac{r(\mathbf{x}_0)}{\beta}\right) p_\theta(\mathbf{x}_0|\mathbf{x}_t) \mathrm{d}\mathbf{x}_0\right)$$

$$\approx \beta \log \left(\exp\left(\frac{r(\mathbf{x}_{0|t})}{\beta}\right)\right) = r(\mathbf{x}_{0|t}). \tag{19}$$

## B  Choice of Diffusion Coefficient

Ma *et al.* [38] have shown that the diffusion coefficient can be chosen freely within the stochastic interpolant framework [1]. Here, we present a more comprehensive proof. We use $\mathbf{w}$ interchangeably to denote the standard Wiener process for both forward and reverse time flows.

**Proposition 1.** *For a linear stochastic process $\mathbf{x}_t = \alpha_t \mathbf{x}_0 + \sigma_t \mathbf{x}_1$ and the Probability-Flow ODE $\mathrm{d}\mathbf{x}_t = u_t(\mathbf{x}_t)\mathrm{d}t$ that yields the marginal density $p_t(\mathbf{x}_t)$, the following forward and reverse SDEs*

*with an arbitrary diffusion coefficient $g_t \geq 0$ share the same marginal density:*

$$\text{Forward SDE: } d\mathbf{x}_t = \left[ u_t(\mathbf{x}_t) + \frac{g_t^2}{2} \nabla \log p_t(\mathbf{x}_t) \right] dt + g_t d\mathbf{w} \tag{20}$$

$$\text{Reverse SDE: } d\mathbf{x}_t = \left[ u_t(\mathbf{x}_t) - \frac{g_t^2}{2} \nabla \log p_t(\mathbf{x}_t) \right] dt + g_t d\mathbf{w}. \tag{21}$$

*Proof.* When velocity field $u_t$ generates a probability density path $p_t$, it satisfies the continuity equation:

$$\frac{\partial}{\partial t} p_t(\mathbf{x}_t) = -\nabla \cdot (p_t(\mathbf{x}_t) u_t(\mathbf{x}_t)). \tag{22}$$

Similarly, for the SDE $d\mathbf{x}_t = \mathbf{f}_t(\mathbf{x}_t) dt + g_t d\mathbf{w}$, the Fokker-Planck equation describes the time evolution of $\tilde{p}_t$:

$$\frac{\partial}{\partial t} \tilde{p}_t(\mathbf{x}_t) = -\nabla \cdot (\tilde{p}_t(\mathbf{x}_t) \mathbf{f}_t(\mathbf{x}_t)) + \frac{1}{2} g_t^2 \nabla^2 \tilde{p}_t(\mathbf{x}_t) \tag{23}$$

where $\nabla^2$ denotes the Laplace operator.

To find an SDE that yields the same marginal probability density as the ODE, we equate the probability density functions in Eq. 23 and Eq. 22, resulting in the following equation:

$$-\nabla \cdot (p_t(\mathbf{x}_t) \mathbf{f}_t(\mathbf{x}_t)) + \frac{1}{2} g_t^2 \nabla^2 p_t(\mathbf{x}_t) = -\nabla \cdot (p_t(\mathbf{x}_t) u_t(\mathbf{x}_t))$$

$$\nabla \cdot (p_t(\mathbf{x}_t)(\mathbf{f}_t(\mathbf{x}_t) - u_t(\mathbf{x}_t))) = \frac{1}{2} g_t^2 \nabla^2 p_t(\mathbf{x}_t) \tag{24}$$

This implies that any SDE with drift coefficient $\mathbf{f}_t(\mathbf{x}_t)$ and diffusion coefficient $g_t$ that satisfies Eq. 24 will generate $p_t$. One particular choice is to set $p_t(\mathbf{x}_t)(\mathbf{f}_t(\mathbf{x}_t) - u_t(\mathbf{x}_t))$ proportional to $\nabla p_t(\mathbf{x}_t)$, i.e., $p_t(\mathbf{x}_t)(\mathbf{f}_t(\mathbf{x}_t) - u_t(\mathbf{x}_t)) = A_t \nabla p_t(\mathbf{x}_t)$. Then Eq. 24 can be rewritten as:

$$A_t \nabla^2 p_t(\mathbf{x}_t) = \frac{1}{2} g_t^2 \nabla^2 p_t(\mathbf{x}_t),$$

which leads to the relation $A_t = \frac{1}{2} g_t^2$. Similarly, the drift coefficient is given by:

$$\mathbf{f}_t(\mathbf{x}_t) = u_t(\mathbf{x}_t) + \frac{1}{2} g_t^2 \frac{\nabla p_t(\mathbf{x}_t)}{p_t(\mathbf{x}_t)}$$

$$= u_t(\mathbf{x}_t) + \frac{1}{2} g_t^2 \nabla \log p_t(\mathbf{x}_t)$$

Thus, a family of SDEs that generate $p_t$ takes the following form:

$$d\mathbf{x}_t = \left[ u_t(\mathbf{x}_t) + \frac{1}{2} g_t^2 \nabla \log p_t(\mathbf{x}_t) \right] dt + g_t d\mathbf{w},$$

which is the forward SDE presented in Eq. 20. Similarly, the reverse SDE in Eq. 21 can be derived by applying the time reversal formula, following Anderson *et al.* [2]. □

**Corollary 1.** *If diffusion coefficient is chosen as $g_t = \sqrt{2 \left( \sigma_t \dot{\sigma}_t - \sigma_t^2 \frac{\dot{\alpha}_t}{\alpha_t} \right)}$ then the score function $\nabla \log p_t(\mathbf{x}_t)$ inside the forward SDE vanish and it can be written as:*

$$d\mathbf{x}_t = \frac{\dot{\alpha}_t}{\alpha_t} \mathbf{x}_t dt + \sqrt{2 \left( \sigma_t \dot{\sigma}_t - \sigma_t^2 \frac{\dot{\alpha}_t}{\alpha_t} \right)} d\mathbf{w} \tag{25}$$

*Proof.* Velocity field $u_t(\mathbf{x}_t)$ for linear stochastic process $\mathbf{X}_t = \alpha_t \mathbf{X}_0 + \sigma_t \mathbf{X}_1$ is given as:

$$u_t(\mathbf{x}_t) = \frac{\dot{\alpha}_t}{\alpha_t} \mathbf{x}_t - \left( \sigma_t \dot{\sigma}_t - \sigma_t^2 \frac{\dot{\alpha}_t}{\alpha_t} \right) \nabla \log p_t(\mathbf{x}_t) \tag{26}$$

Plugging this equation into forward SDE Eq. 20, we can immediately see that when $g_t = \sqrt{2 \left( \sigma_t \dot{\sigma}_t - \sigma_t^2 \frac{\dot{\alpha}_t}{\alpha_t} \right)}$ the score function term vanishes and the remaining terms constitute Eq. 25. □

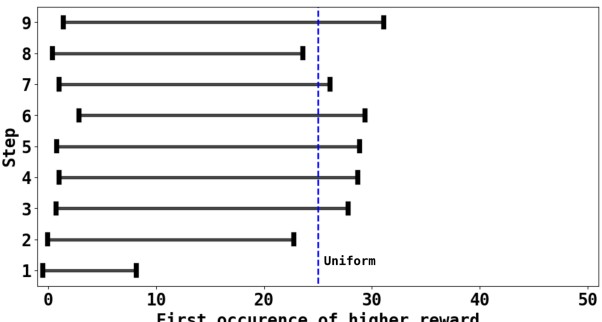

Figure 9: **Analysis of number of function evaluations (NFEs) across timesteps.** The NFEs required to achieve a higher reward for each timestep. The plot illustrates the $\pm 1$ sigma variation band. The blue-dotted line represents the uniform allocation of compute (NFEs) across timesteps. We observe that the NFEs required to identify a higher-reward sample may exceed the uniformly allocated budget (blue dotted line).

## C  Adaptive Time Scheduling and Rollover Strategy

In this section, we provide details of adaptive time scheduling and NFE analysis result which inspired rollover strategy.

**Adaptive Time Scheduling.**  As discussed in Sec. 4.3, to maximize the exploration space in VP-SDE sampling, we design the time scheduler to take smaller steps during the initial phase—when variance is high—and gradually increase the step size in later stages. Specifically, we define the time scheduler as $t_{\text{new}} = \sqrt{1 - (1 - t)^2}$. While this approach can be problematic when the number of steps is too low—resulting in excessively large discretization steps in later iterations—we find that using a reasonable number of steps (e.g.,10) works well in practice, benefiting from the few-step generation capability of flow models. This setup effectively balances a broad exploration space with fast inference time, highlighting one of the key advantages of flow models over diffusion models.

**NFE Analysis.**  As discussed in Sec. 6, we analyze the number of function evaluations (NFEs) required to obtain a sample with a higher reward than the current one. In Fig. 9, we visualize the variance band of the required NFEs across timesteps, with the blue-dotted line representing the uniform allocation used in previous particle sampling methods [30, 53]. Notably, uniform compute allocation may constrain exploration and fail to identify high-reward samples, as evidenced by crossings within the variance band. This observation motivates the use of a rollover strategy to optimize compute utilization efficiently. As demonstrated in Sec. 7, our experiments confirm that RBF provides additional improvements over previous particle sampling methods [30, 53].

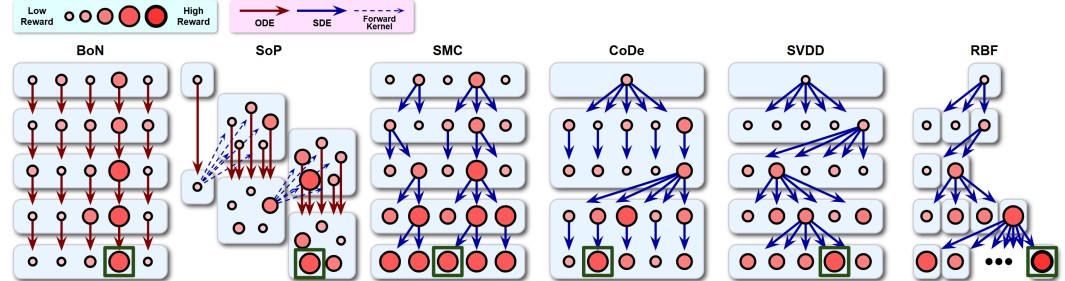

Figure 10: **Schematics of inference-time search algorithms.** Linear-ODE-based methods, BoN and SoP use a deterministic sampling process, whereas particle-sampling-based methods follow a stochastic process. Note that RBF adaptively allocates NFEs across denoising timesteps.

## D  Search Algorithms

In this section, we introduce the inference-time search algorithms discussed in Sec. 2 along with their implementation details. An illustrative figure of the algorithms is provided in Fig. 10. Here, we

define the batch size $(N)$ as the number of initial latent samples and the particle size $(K)$ as the number of samples drawn from the proposal distribution $p_\theta(\mathbf{x}_{t-\Delta t}|\mathbf{x}_t)$ at each denoising step.

**Best-of-N** (BoN) [57, 58] is a form of rejection sampling. Given $N$ generated samples $\{\mathbf{x}_0^{(i)}\}_{i=1}^N$, BoN selects the sample with the highest reward.

$$\mathbf{x}_0 = \underset{\{\mathbf{x}_0^{(i)}\}_{i=1}^N}{\arg\max}\, r(\mathbf{x}_0^{(i)}).$$

As presented in Sec. 7, we fixed the total compute budget to 500 NFEs and the number of denoising steps to 10, which sets the batch size of BoN to $N = 50$.

**Search over Paths** (SoP) [39] begins by sampling $N$ initial noises and running the ODE solver up to a predefined timestep $t_0$. Then the following two operations iterate until reaching $t = 0$:

1. Applying the forward kernel: For each sample in the batch at time $t$, $K$ particles are sampled using the forward kernel, which propagates them from $t$ to $t + \Delta_f$.

2. Solving the ODE: The resulting $N \cdot K$ particles are then evolved from $t + \Delta_f$ to $t + \Delta_f - \Delta_b$ by solving the ODE. The top $N$ candidates with the highest rewards are selected.

We followed the original implementations [39] for $\Delta_f$ and $\Delta_b$. We used $N = 2$ and $K = 5$.

**Sequential Monte Carlo** (SMC) [23, 14] extends the idea of importance sampling to a time-sequential setting by maintaining $N$ samples and updating their importance weights over time:

$$w_{t-\Delta t}^{(i)} = \frac{p_\theta^*(\mathbf{x}_{t-\Delta t}|\mathbf{x}_t)}{q(\mathbf{x}_{t-\Delta t}|\mathbf{x}_t)} w_t^{(i)} = \frac{p_\theta(\mathbf{x}_{t-\Delta t}|\mathbf{x}_t)\exp\left(v(\mathbf{x}_{t-\Delta t}^{(i)})/\beta\right)}{q(\mathbf{x}_{t-\Delta t}|\mathbf{x}_t)\exp\left(v(\mathbf{x}_t^{(i)})/\beta\right)} w_t^{(i)},$$

where $q(\mathbf{x}_{t-\Delta t}|\mathbf{x}_t)$ is a proposal distribution and the last equality follows from the optimal policy Eq. 18. We used the reverse process of the pretrained model as the proposal distribution, which leads to the following importance weight equation:

$$w_{t-\Delta t}^{(i)} = \frac{\exp\left(v(\mathbf{x}_{t-\Delta t}^{(i)})/\beta\right)}{\exp\left(v(\mathbf{x}_t^{(i)})/\beta\right)} w_t^{(i)}. \tag{27}$$

At each step when effective sample size $\left(\sum_{j=1}^N w_t^{(j)}\right)^2 / \sum_{i=1}^N (w_t^{(i)})^2$ is below the threshold, we perform resampling, i.e.,, indices $\{a_t^{(i)}\}_{i=1}^N$ are first sampled from a multinomial distribution based on the normalized importance weights:

$$\{a_t^{(i)}\}_{i=1}^N \sim \text{Multinomial}\left(N, \quad \left\{\frac{w_t^{(i)}}{\sum_{j=1}^N w_t^{(j)}}\right\}_{i=1}^N\right).$$

These ancestor indices $a_t^{(i)}$ are then used to replicate high-weight particles and discard low-weight ones, yielding the resampled set $\{\mathbf{x}_t^{(a_t^{(i)})}\}_{i=1}^N$. If resampling is not performed, the indices are simply set as $a_t^{(i)} = i$. Lastly, one-step denoised samples are obtained from $\{\mathbf{x}_t^{(a_t^{(i)})}\}_{i=1}^N$:

$$\mathbf{x}_{t-\Delta t}^{(i)} \sim p_\theta(\mathbf{x}_{t-\Delta t}|\mathbf{x}_t^{(a_t^{(i)})}).$$

When resampling is performed, the importance weights are reinitialized to one, i.e.,, $w_t = \mathbf{1}$. The importance weights for the next step, $w_{t-\Delta t}$ are subsequently computed according to Eq. 27, regardless of whether resampling was applied.
We used $N = 50$ for all applications.

**Controlled Denoising** (CoDe) [53] extends BoN by incorporating an interleaved selection step after every $L$ denoising steps.

$$\mathbf{x}_{t-L\Delta t} = \underset{\{\mathbf{x}_{t-L\Delta t}^{(i)}\}_{i=1}^{K}}{\arg\max} \; \exp\left(v(\mathbf{x}_{t-L\Delta t}^{(i)})/\beta\right)$$

We used $N = 2$, $K = 25$, and $L = 2$ for all applications.

**SVDD** [30] approximates the optimal policy in Eq. 3 by leveraging weighted $K$ particles:

$$p_\theta^*(\mathbf{x}_{t-\Delta t}|\mathbf{x}_t) \approx \sum_{i=1}^{K} \frac{w_{t-\Delta t}^{(i)}}{\sum_{j=1}^{K} w_{t-\Delta t}^{(j)}} \delta_{\mathbf{x}_{t-\Delta t}^{(i)}} \tag{28}$$

$$\{\mathbf{x}_{t-\Delta t}^{(i)}\}_{i=1}^{K} \sim p_\theta(\mathbf{x}_{t-\Delta t}|\mathbf{x}_t)$$

$$w_{t-\Delta t}^{(i)} = \exp\left(v(\mathbf{x}_{t-\Delta t}^{(i)})/\beta\right).$$

At each timestep, the approximate optimal policy in Eq. 28 is sampled by first drawing an index $a_{t-\Delta t}$ from a categorical distribution:

$$a_{t-\Delta t} \sim \text{Categorical}\left(\left\{\frac{w_{t-\Delta t}^{(i)}}{\sum_{j=1}^{K} w_{t-\Delta t}^{(j)}}\right\}_{i=1}^{K}\right) \tag{29}$$

This index is then used to select the sample from $\{\mathbf{x}_{t-\Delta t}^{(i)}\}_{i=1}^{K}$, i.e., $\mathbf{x}_{t-\Delta t} \leftarrow \mathbf{x}_{t-\Delta t}^{(a_{t-\Delta t})}$. In practice, SVDD uses $\beta = 0$, replacing sampling from the categorical distribution with a direct $\arg\max$ operation, i.e., selecting the particle with the largest importance weight. Following the original implementation [30], we used $N = 2$ and $K = 25$ for all applications.

**Rollover Budget Forcing** (RBF) adaptively allocates compute across denoising timesteps. At each timestep, when a particle with a higher reward than the previous one is discovered, it immediately takes a denoising step, and the remaining NFEs are rolled over to the next timestep, ensuring efficient utilization of the available compute. To maintain consistency with SVDD [30], we set $N = 2$, with the compute initially allocated uniformly across all timesteps. We present the pseudocode for sampling from the stochastic proposal distribution with interpolant conversion in Alg. 1. Specifically, the pseudocode for RBF with SDE conversion and interpolant conversion is provided in Alg. 2. Here, we denote $\{S^{(i)}\}_{i=1}^{M}$ as a sequence of timesteps in descending order, where $S^{(1)} = 1$ and $S^{(M)} = 0$, and $M$ is the total number of denoising steps.

## E    Additional Results

### E.1    Aesthetic Image Generation

In this section, we demonstrate that inference-time scaling can also be applied to gradient-based methods, such as DPS [8], for differentiable rewards. Specifically, we consider aesthetic image generation and show that RBF leads to synergistic performance improvements. We first derive the formulation of the proposal distribution for differentiable rewards and then present qualitative and quantitative results.

### E.1.1    Gradient-Based Guidance

Uehara *et al.* [61] have shown that the marginal distribution $p_t^*(\mathbf{x}_t)$ is computed as follows:

$$p_t^*(\mathbf{x}_t) \propto \exp\left(\frac{v(\mathbf{x}_t)}{\beta}\right) p_t(\mathbf{x}_t) \approx \exp\left(\frac{r(\mathbf{x}_{0|t})}{\beta}\right) p_t(\mathbf{x}_t),$$

**Algorithm 1:** `stoch_denoise`: 1-step stochastic denoising

**Inputs:** original velocity field $u$,

original interpolant $(\alpha, \sigma)$,

new interpolant $(\bar{\alpha}, \bar{\sigma})$,

diffusion coefficient $g$, current

sample $\bar{\mathbf{x}}_s$, current timestep $s$,

denoising step size $\Delta s$

**Outputs:** Stochastically denoised

sample $\bar{\mathbf{x}}_{s-\Delta s}$

1   $t_s \leftarrow \rho^{-1}(\bar{\rho}(s))$    $c_s \leftarrow \bar{\sigma}_s/\sigma_{t_s}$

2   $\bar{\mathbf{u}}_s \leftarrow \frac{\dot{c}_s}{c_s}\bar{\mathbf{x}}_s + c_s\dot{t}_s u_{t_s}\left(\frac{\bar{\mathbf{x}}_s}{c_s}\right)$   // Eq. 11

3   $\mathbf{s}_s \leftarrow \frac{1}{\bar{\sigma}_s}\frac{\bar{\alpha}_s\bar{\mathbf{u}}_s-\dot{\bar{\alpha}}_s\bar{\mathbf{x}}_s}{\bar{\alpha}_s\bar{\sigma}_s-\bar{\alpha}_s\dot{\bar{\sigma}}_s}$      // Eq. 8

4   $\mathbf{f}_s = \bar{\mathbf{u}}_s - \frac{g_s^2}{2}\mathbf{s}_s$      // Eq. 7

5   $\mathbf{z} \sim \mathcal{N}(\mathbf{0}, \boldsymbol{I})$

6   $\bar{\mathbf{x}}_{s-\Delta s} \leftarrow \bar{\mathbf{x}}_s - \mathbf{f}_s\Delta s + g_s\sqrt{\Delta s}\,\mathbf{z}$

---

**Algorithm 2:** Rollover Budget Forcing (RBF)

**Inputs:** Number of denoising steps $M$,

timesteps $\{S^{(i)}\}_{i=1}^M$, NFE quota

$\{Q^{(i)}\}_{i=1}^M$

**Outputs:** Aligned sample $\bar{\mathbf{x}}_0$

1   $\bar{\mathbf{x}}_1 \sim \mathcal{N}(0, \boldsymbol{I})$    $r^* \leftarrow r(\bar{\mathbf{x}}_{0|1})$

2   **for** $i \in \{1, \ldots, M\}$ **do**

3     $s \leftarrow S^{(i)}$    $\Delta s \leftarrow S^{(i)} - S^{(i+1)}$    $q \leftarrow Q^{(i)}$

4     **for** $j \in \{1, \ldots, q\}$ **do**

5       $\bar{\mathbf{x}}_{s-\Delta s}^{(j)} \leftarrow$ `stoch_denoise`$(\bar{\mathbf{x}}_s, s, \Delta s)$

       // Alg. 1

6       **if** $r^* < r(\bar{\mathbf{x}}_{0|s-\Delta s}^{(j)})$ **then**

7         $Q^{(i+1)} \leftarrow Q^{(i+1)} + Q^{(i)} - j$

         // Sec. 6

8         $r^* \leftarrow r(\bar{\mathbf{x}}_{0|s-\Delta s}^{(j)})$    $\bar{\mathbf{x}}_{s-\Delta s} \leftarrow$

        $\bar{\mathbf{x}}_{s-\Delta s}^{(j)}$

9         **break**

10      **if** $j = q$ **then**

11        $k^* \leftarrow \arg\max_{k \in \{1, \ldots, q\}} r(\bar{\mathbf{x}}_{0|s-\Delta s}^{(k)})$

12        $\bar{\mathbf{x}}_{s-\Delta s} \leftarrow \bar{\mathbf{x}}_{s-\Delta s}^{(k^*)}$

---

|  | FLUX [28] | DPS [8] | SVDD [30] + DPS [8] | RBF (Ours) + DPS [8] |

*"Bird"*

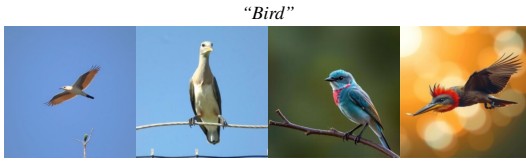

*"Bat"*

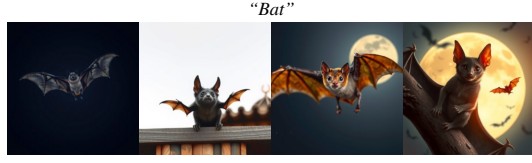

Figure 11: **Qualitative results of aesthetic image generation.** At inference-time, we guide generate using the aesthetic score [50] as the given reward, which assesses **visual appeal**.

Table 2: **Quantitative results of aesthetic image generation.** $^\dagger$ denotes the given reward used in inference time. The best result in each row is highlighted in **bold**.

| Model | Aesthetic Score$^\dagger$ [50] | ImageReward [67] (held-out) |
|---|---|---|
| FLUX [28] | 5.795 | 0.991 |
| DPS [8] | 6.438 | 0.605 |
| SVDD [30]+DPS [8] | 6.887 | 1.077 |
| RBF (Ours)+DPS [8] | **7.170** | **1.152** |

where the approximation follows from Eq. 19. When the reward is differentiable (e.g.,, aesthetic score [50]), one can simulate samples from $p_t^*(\mathbf{x}_t)$ by computing its score function:

$$
\nabla \log p_t^*(\mathbf{x}_t) = \nabla \log \left[\exp\left(\frac{r(\mathbf{x}_{0|t})}{\beta}\right)p_t(\mathbf{x}_t)\right]
$$

$$
= \frac{1}{\beta}\underbrace{\nabla r(\mathbf{x}_{0|t})}_{\text{Guidance}} + \underbrace{\nabla \log p_t(\mathbf{x}_t)}_{\text{Pretrained Score}}. \tag{30}
$$

For differentiable rewards, we incorporate the gradient-based guidance defined in Eq. 30 into the SDE sampling process described in Eq. 7. Notably, this approach is orthogonal to inference-time

Table 3: **Comparison of diffusion and flow models**.

| Type | Model | ImageReward [67] | HPS [65] | PickScore [26] | CLIP Score [44] | Steps |
|------|-------|------------------|----------|----------------|-----------------|-------|
| Diffusion | SD2 [48] | 0.429 | 0.280 | 0.218 | 0.269 | 50 |
| | SANA-1.5 [66] | 0.894 | 0.284 | 0.222 | 0.270 | 20 |
| Flow | SD3 [15] | 1.154 | 0.294 | 0.226 | 0.277 | 28 |
| | FLUX [28] | 1.054 | 0.290 | 0.226 | 0.275 | 5 |

scaling, and RBF can be additionally utilized to further enhance performance. In the next section, we experimentally demonstrate that RBF can be effectively integrated with gradient-based guidance.

### E.1.2 Aesthetic Image Generation Results

The aesthetic image generation task aims to sample images that best capture human preferences, such as visual appeal. We use $45$ animal prompts from previous work, DDPO [5]. The aesthetic score [50] serves as the given reward, while ImageReward [67] is used as the held-out reward.

We present quantitative and qualitative results of aesthetic image generation in Tab. 2 and Fig. 11. Notably, RBF, implemented with DPS [8], achieves significant improvements on both the given and held-out rewards, even surpassing SVDD [30]. Qualitatively, RBF effectively adapts the pretrained flow model to better align with human preferences, particularly in terms of visual appeal.

### E.2 Comparison of Diffusion and Flow Models

We present quantitative comparisons between text-to-image diffusion and flow models in Tab. 3, using compositional text prompts from GenAI-Bench [21]. As shown, flow-based models outperform diffusion models across all evaluation metrics assessing image quality [67, 65, 26] and text alignment [44, 67]. In the flow-based models, FLUX [28] achieves competitive performance while requiring fewer steps compared to Stable Diffusion 3 [15].

### E.3 Scaling Behavior Comparison

As discussed in Sec. 4, expanding the exploration space and applying budget forcing significantly enhance the efficiency of RBF, leading to superior performance improvements over BoN. Here, we compare the scaling behavior of BoN, a representative Linear-ODE-based method, with RBF across different numbers of function evaluations (NFEs).

We report qualitative and quantitative scaling results for quantity-aware image generation (Fig. 12, Tab. 4) and for compositional text-to-image generation (Fig. 13, Tab. 5), respectively. Our results indicate that allocating more compute leads to performance improvements for both BoN and RBF. However, the performance of BoN plateaus after 300 NFEs, whereas RBF continues to scale and achieves the highest reward in both tasks. Notably, RBF shows similar trend in the held-out reward, outperforming BoN and demonstrating its efficiency.

**Time Complexity and Compute Analysis.** We present time complexity of scaling methods in Tab. 6. Let $S$ as the number of denoising steps, $N$ as the NFE budget, and $c_s$ and $c_v$ as the costs of the denoising and verification, respectively. Since all methods share the same NFE budget $N$, the total denoising cost is fixed at $N \cdot c_d$. For the verification cost, although BoN has the lowest cost, RBF consistently outperforms BoN across all NFE budget regimes in both compositional text-to-image generation and quantity-aware image generation tasks (Fig. 12 and Fig. 13) while incurring only a marginal increase in verification overhead.

Additionally, at inference time, a user can specify the compute budget (NFEs), which determines the total runtime of our method. We report the runtime of RBF in Tab. 7. Under a 500-NFE budget, scaling for compositional text-to-image generation (VQAScore [31]) requires 635.01 seconds per image. Runtime can be reduced by lowering the NFE budget—at the cost of reward performance—and further accelerated by decreasing output resolution or increasing batch size.

Table 4: **Quantitative results of quantity-aware image generation in NFE scaling expriment.** We use the same 100 prompts from T2I-CompBench [20]. $^\dagger$ denotes the given reward.

| | NFEs | RSS$^\dagger$ [34] ↓ | Acc. ↑ | VQAScore [31] (held-out) ↑ | Aesthetic Score [50] ↑ |
|---|---|---|---|---|---|
| BoN | 50 | 4.360 | 0.400 | 0.758 | 5.408 |
| | 100 | 3.280 | 0.510 | 0.750 | 5.522 |
| | 300 | 2.190 | 0.570 | 0.755 | 5.463 |
| | 500 | 1.760 | 0.580 | 0.756 | 5.420 |
| | 1000 | 1.340 | 0.590 | 0.759 | 5.466 |
| RBF (Ours) | 50 | 3.250 | 0.410 | 0.756 | 5.560 |
| | 100 | 1.860 | 0.590 | 0.760 | 5.627 |
| | 300 | 0.690 | 0.720 | 0.779 | 5.503 |
| | 500 | 0.540 | 0.800 | 0.769 | 5.581 |
| | 1000 | 0.290 | 0.880 | 0.777 | 5.526 |

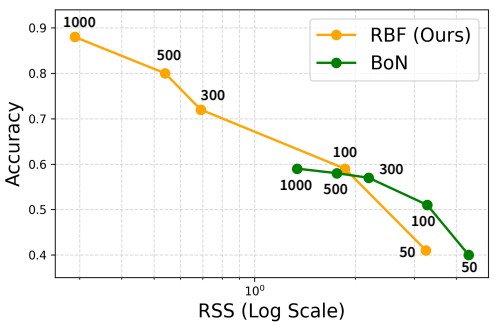

Figure 12: **Quantity-aware image generation scaling behavior comparison of BoN and** RBF**.** We plot the known reward (RSS) [34] against accuracy for different numbers of function evaluations: $\{50, 100, 300, 500, 1,000\}$. Note that the horizontal axis is displayed on a logarithmic scale.

Table 5: **Quantitative results of compositional text-to-image generation in NFE scaling expriment.** We use the 121 prompts from GenAI-Bench [21]. $^\dagger$ denotes the given reward.

| | NFEs | VQAScore$^\dagger$ [31] ↑ | Inst.BLIP [10] (held-out) ↑ | Aesthetic [50] ↑ |
|---|---|---|---|---|
| BoN | 50 | 0.8310 | 0.8011 | 5.2246 |
| | 100 | 0.8459 | 0.7959 | 5.2594 |
| | 300 | 0.8775 | 0.8250 | 5.1414 |
| | 500 | 0.8790 | 0.8200 | 5.1620 |
| | 1000 | 0.8886 | 0.8269 | 5.2055 |
| RBF (Ours) | 50 | 0.8577 | 0.8253 | 5.2704 |
| | 100 | 0.8824 | 0.8212 | 5.3213 |
| | 300 | 0.9146 | 0.8387 | 5.2837 |
| | 500 | 0.9250 | 0.8430 | 5.2370 |
| | 1000 | 0.9283 | 0.8369 | 5.2593 |

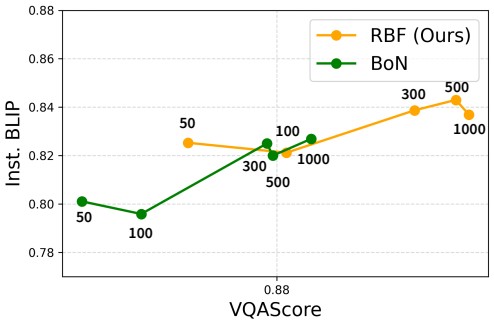

Figure 13: **Compositional text-to-image generation scaling behavior comparison of BoN and** RBF**.** We plot the known reward (VQAScore) [31] against the held-out reward [10] for different numbers of function evaluations: $\{50, 100, 300, 500, 1,000\}$.

Table 6: **Time complexity of scaling methods.**

| Base | BoN | SMC [23] | SVDD [30], RBF |
|---|---|---|---|
| $S \cdot c_d$ | $N \cdot c_d + \frac{N}{S} \cdot c_v$ | $N \cdot c_d + N \cdot c_v$ | $N \cdot c_d + N \cdot c_v$ |

Table 7: **Runtime of** RBF**.**

| | 50 | 100 | 300 | 500 | 1000 |
|---|---|---|---|---|---|
| Runtime (sec) | 84.00 | 140.11 | 383.89 | 635.01 | 1243.68 |
| VQAScore [31] | 0.858 | 0.882 | 0.915 | 0.925 | 0.928 |

For all experiments, we use FLUX [28], which requires approximately 32GB of GPU memory, accounting for the majority of overall memory usage. All evaluations are performed on an NVIDIA RTX A6000 GPU.

# F Implementation Details

## F.1 Choice of Hyperparameters

We report quantitative results on aesthetic score [50] and diversity [23] for images generated under different settings of the number of denoising steps and the diffusion coefficient. As shown in Tab. 8(a), the number of denoising steps beyond 10 gives marginal gains. Hence, we fixed the number of denoising steps to 10 to ensure fair and efficient evaluation across all methods. Note that once the

Table 8: **Choice of hyperparameters.** Evaluation of the images generated with different (a) number of denoising steps and (b) diffusion coefficient.

| Steps | Aesthetic [50] | Diversity [23] |
|---|---|---|
| 10 | **5.635** | **0.084** |
| 20 | 5.680 | 0.103 |

(a) Number of denoising steps

| Norm | Aesthetic [50] $g(t) = t$ | Diversity [23] $g(t) = t$ | Aesthetic [50] $g(t) = t^2$ | Diversity [23] $g(t) = t^2$ |
|---|---|---|---|---|
| 1 | 5.635 | 0.084 | 5.652 | 0.083 |
| 3 | 5.168 | 0.153 | **5.436** | **0.158** |
| 5 | 4.608 | 0.223 | 4.838 | 0.187 |

(b) Diffusion coefficient

number of denoising steps is fixed, the total particle count per step is automatically determined by dividing the total NFE budget by the number of steps. Additionally, Tab. 8(b) reports results obtained under varying diffusion coefficients scaled by different norms. We found that using $g(t) = 3t^2$ consistently offered the best trade-off between sample diversity and output fidelity, so we adopt this setting for all SDE sampling.

## F.2 Compositional Text-to-Image Generation

In the compositional text-to-image generation task, we use the VQAScore as the reward, which evaluates image-text alignment using a visual question-answering (VQA) model (CLIP-FlanT5 [31] and InstructBLIP [10]). Specifically, VQAScore measures the probability that a given attribute or object is present in the generated image. To compute the reward, we scale the probability value by setting $\beta = 0.1$ in Eq. 3.

## F.3 Quantity-Aware Image Generation

In quantity-aware image generation, text prompts specify objects along with their respective quantities. To generate images that accurately match the specified object counts, we use the negation of the Residual Sum of Squares (RSS) as the given reward. Here, RSS is computed to measure the discrepancy between the detected object count $\hat{C}_i$ and the target object count $C_i$ in the text prompt: $\text{RSS} = \sum_{i=1}^{n} \left( C_i - \hat{C}_i \right)^2$, where $n$ is the total number of object categories in the prompt. We additionally report accuracy, which is defined as 1 when RSS = 0 and 0 otherwise. For the held-out reward, we report VQAScore measured with CLIP-FlanT5 [31] model.

**Object Detection Implementation Details.** To compute the given reward, RSS, it is necessary to detect the number of objects per category, $\hat{C}_i$. Here, we leverage the state-of-the-art object detection model, GroundingDINO [34] and the object segmentation model SAM [25], which is specifically used to filter out duplicate detections.

We observe that naïvely using the detection model [34] to compute RSS leads to poor detection accuracy due to two key issues: inner-class duplication and cross-class duplication. Inner-class duplication occurs when multiple detections are assigned to the same object within a category, leading to overcounting. This often happens when an object is detected both individually and as part of a larger group. Cross-class duplication arises when an object is assigned to multiple categories due to shared characteristics (e.g.,, a toy airplane being classified as both a toy and an airplane), making it difficult to assign it to a single category.

To address inner-class duplication, we refine the object bounding boxes detected by GroundingDINO [34] using SAM [25] and filter out overlapping detections. Smaller bounding boxes are prioritized, and larger ones that significantly overlap with existing detections are discarded. This ensures that each object is counted only once within its category. To resolve cross-class duplication, we assign each object to the category with the highest GroundingDINO [34] confidence score which prevents duplicate counting across multiple classes.

**More qualitative results are presented in the following pages.**

# G    Additional Qualitative Results

## G.1    Comparisons of Inference-Time SDE Conversion and Interpolant Conversion

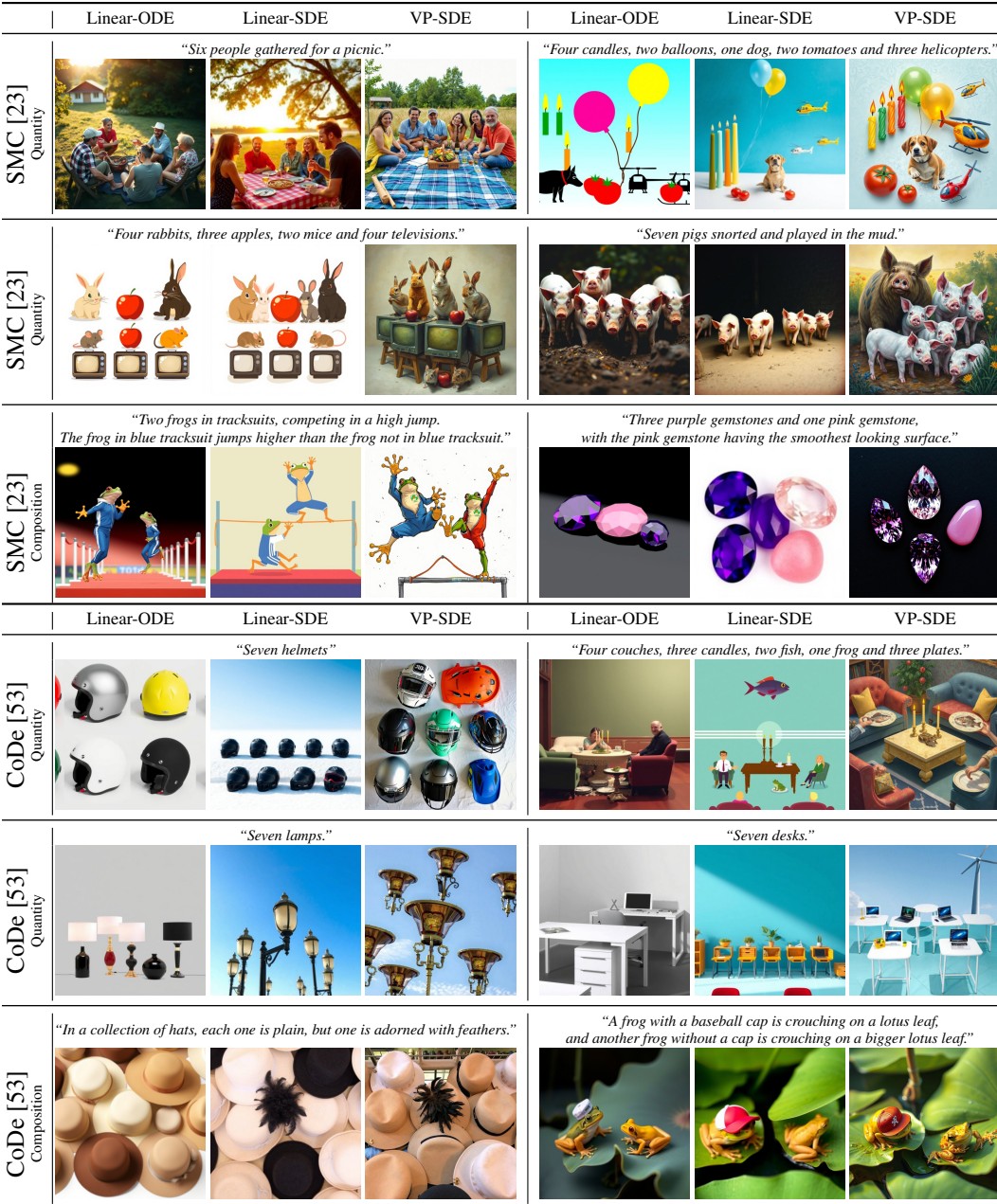

Figure 14: **Additional qualitative results of inference-time SDE conversion and interpolant conversion.**

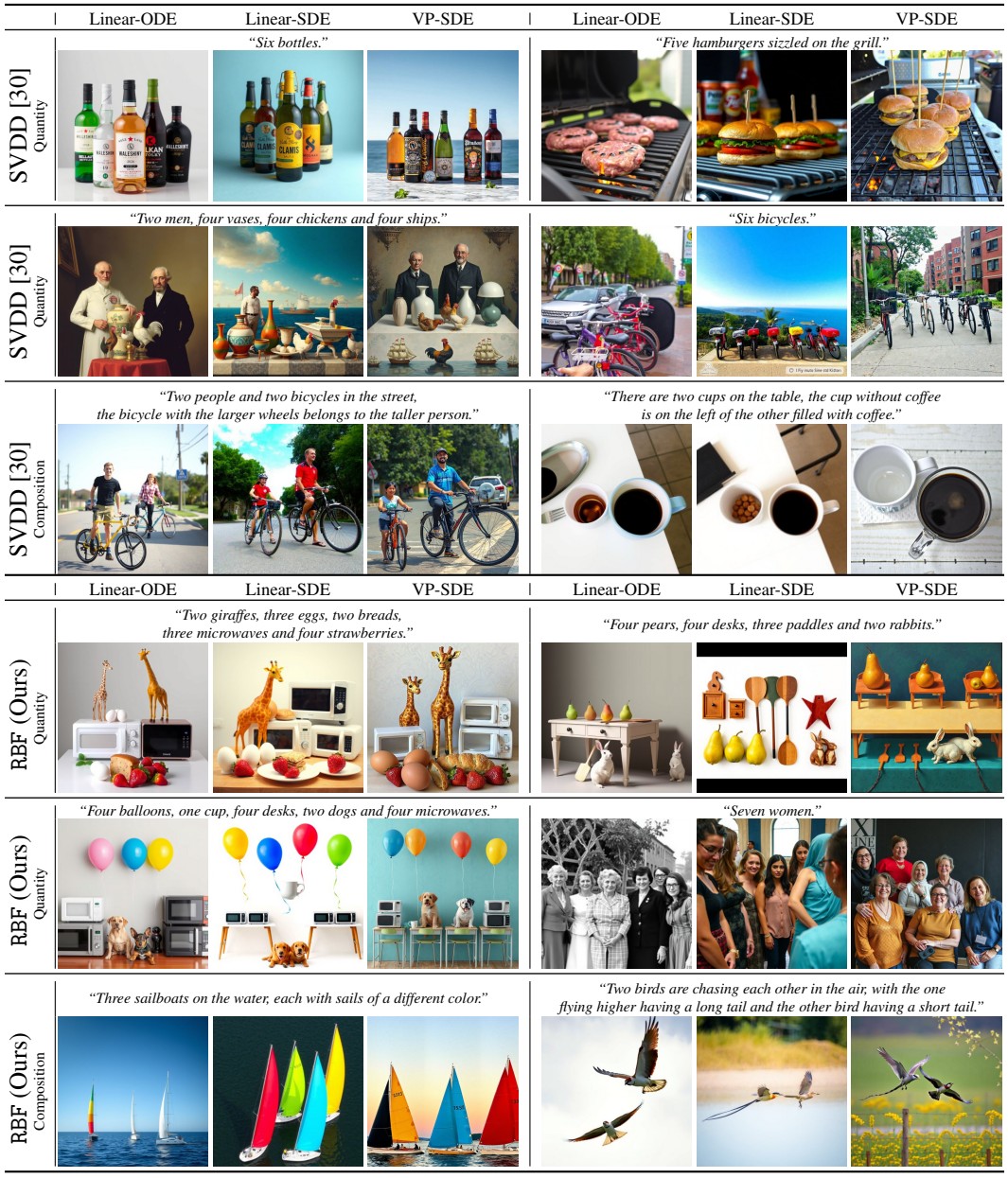

Figure 15: **Additional qualitative results of inference-time SDE conversion and interpolant conversion.**

## G.2 Comparisons of Inference-Time Scaling

| BoN | SoP [39] | SMC [23] | CoDe [53] | SVDD [30] | RBF (Ours) |
|-----|----------|----------|-----------|-----------|------------|

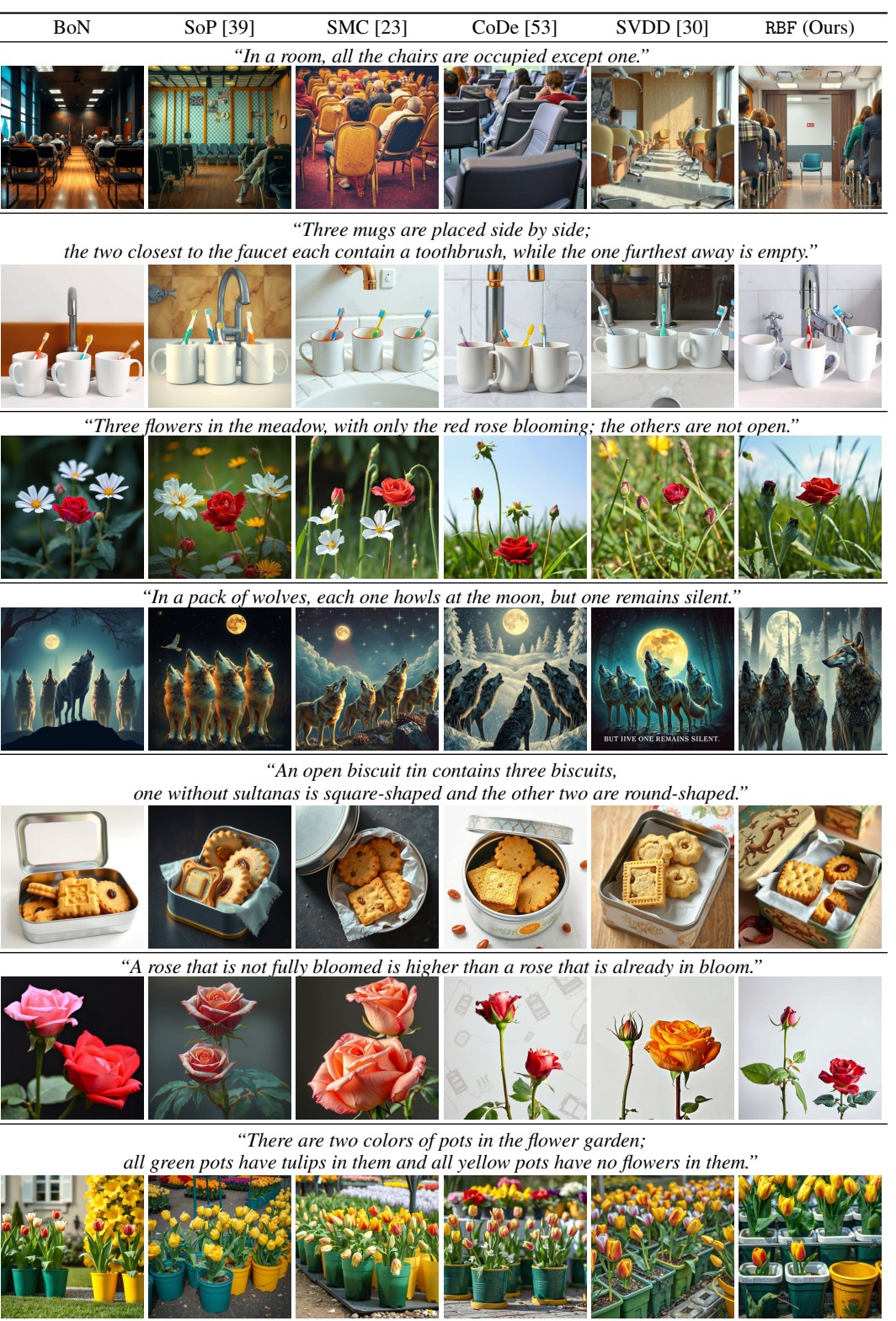

Figure 16: **Additional qualitative results of compositional text-to-image generation task**.

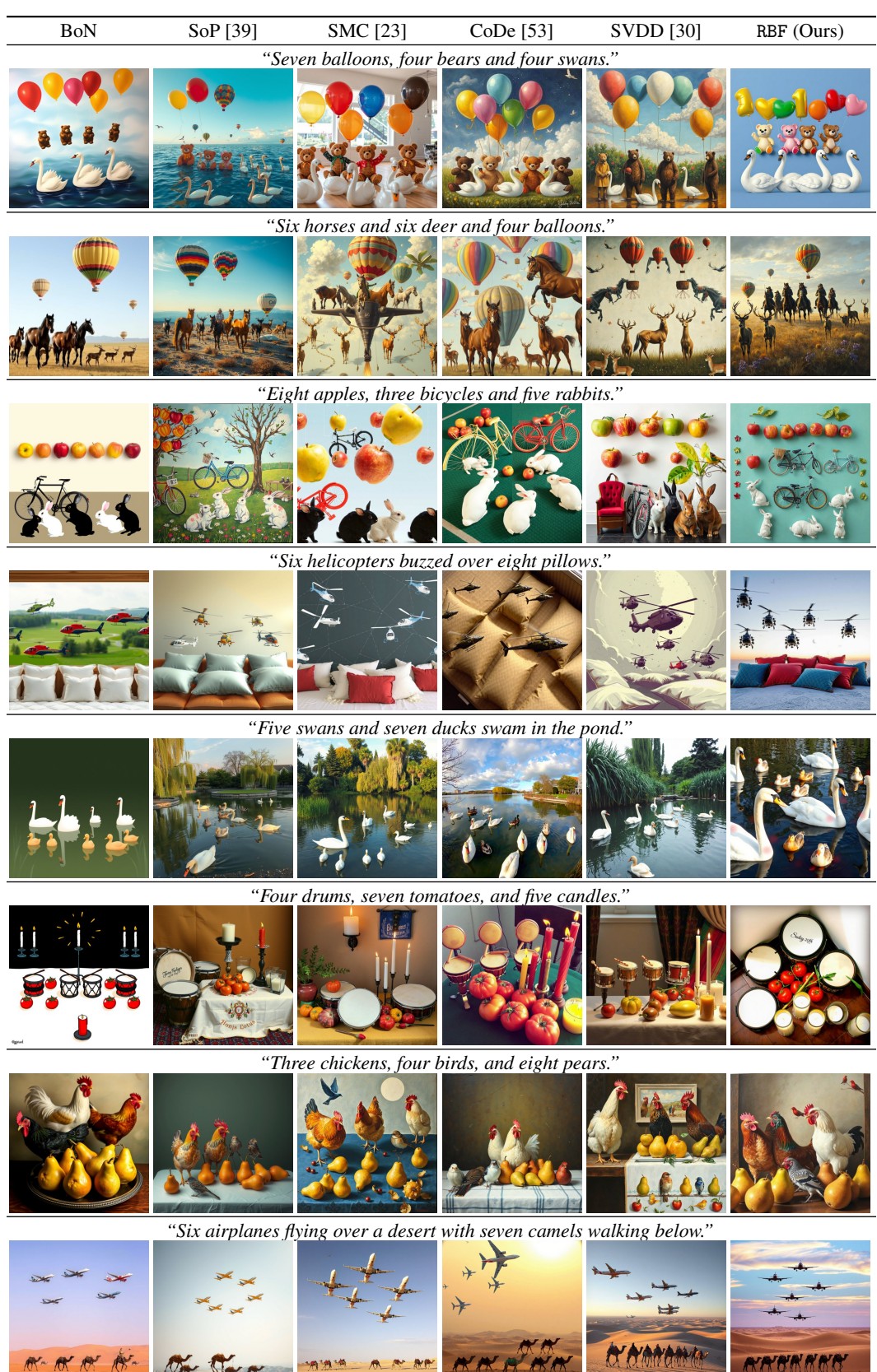

Figure 17: **Additional qualitative results of quantity-aware image generation task**.

