# OpenReview forum: "Inference-Time Scaling for Flow Models via Stochastic Generation and Rollover Budget Forcing"
_NeurIPS.cc/2025/Conference — NeurIPS 2025 poster_

### Official Review · Reviewer_Hs6F · 2025-06-11

**Clarity:** 4
**Significance:** 2
**Originality:** 2
**Rating:** 4
**Confidence:** 3

**Summary:**

The paper proposes a new method for inference-time scaling in pretrained flow models, making them competitive with diffusion models for tasks requiring reward alignment (e.g., compositional or quantity-aware image generation). The key contributions are:  introducing SDE-based stochasticity into flow models for effective particle sampling, interpolant conversion (linear to variance-preserving) to broaden the search space/diversity, and Rollover Budget Forcing (RBF) for adaptive allocation of computational budgets during inference.

**Questions:**

- In Section 5, the authors suggest that an uneven allocation may be more effective than a uniform one. This idea has been previously explored in several works on flow models [1], [2] and diffusion models [3]. In particular, [2] shows that the trajectory of flow models tends to be more curved at the beginning and end of the denoising process, implying that more NFEs should be allocated to these stages. Does this observation support the design presented in Appendix D, or does it contradict it?

- From my perspective, the inference time scaling for the diffusion model is especially valuable in tasks such as molecular generation where the runtime does not matter. It would be great to see how this work is applied to this domain.

**Ethical Concerns:**

["NO or VERY MINOR ethics concerns only"]

**Final Justification:**

The authors have fully addressed my concerns, so I decided to raise the score. However, after reading the rebuttal, I realized that some points are outside my expertise, so I lowered my confidence score.

**Limitations:**

yes

**References**

[1] Shaul, N., Perez, J., Chen, R. T., Thabet, A., Pumarola, A., & Lipman, Y. (2023). Bespoke solvers for generative flow models. arXiv preprint arXiv:2310.19075.

[2] Bao Nguyen, Binh Nguyen, and Viet Anh Nguyen. "Bellman Optimal Step-size Straightening of Flow-Matching Models." arXiv preprint arXiv:2312.16414 (2023).

[3] Lu, Cheng, et al. "Dpm-solver: A fast ode solver for diffusion probabilistic model sampling in around 10 steps." Advances in Neural Information Processing Systems 35 (2022).

**Quality:**

3

**Strengths And Weaknesses:**

**Strengths**:

- Includes diverse test cases, comparisons to multiple baselines, and ablation on key techniques (SDE, interpolant conversion, RBF).
- Outperforms prior inference-time scaling baselines on compositional and quantity-aware image generation tasks. Demonstrates both quantitative and qualitative improvements.
- Clear and easy-to-follow presentation.

**Weaknesses**:

- It is ambiguous why authors choose this setup heuristically: “In our case, we set $g(t) = t^2$, scaled by a factor of 3.”
- After all the transformations, the paper finally uses the VP-SDE. I questioned the novelty of the paper. It appears that all transformations can be eliminated by utilizing direct paths in the VP-SDE diffusion model. This point should be discussed, or at least an experiment around this selection should be included.
- Some papers show that the denoising steps of flow models should not be uniform [1], [2]. A comparison between your time schedule design in Appendix D and the time schedule in these papers is recommended.
- The paper introduces methods that add computational cost at inference. There’s limited discussion or quantitative analysis of this trade-off.
- The paper consists of several modules that are built upon each other. However, the paper lacks some ablation studies on them and important hyperparameters.

---

> ### Author Rebuttal · Authors · 2025-07-30
>
> **[W1: Choice of hyperparameters]**
>
> We clarify our choice of hyperparameters. We set denoising steps to 10 for all experiments and diffusion coefficient $g(t) = 3t^2$. Below we provide rationale of this choice.
>
> **Tab. R1**: *Number of denoising step*
> | Steps | Aesthetic | Diversity |
> |-------|------------------|-----------|
> | 10    | **5.635**            | **0.084**     |
> | 20    | 5.680            | 0.103     |
>
> **Tab. R2**: *Diffusion coefficient*
> | Norm | Aesthetic $g(t)=t$ | Diversity $g(t)=t$ | Aesthetic $g(t)=t^2$ | Diversity $g(t)=t^2$ |
> |------|---------------------------|----------------------------|---------------------|----------------------|
> | 1    | 5.635                     | 0.084                      | 5.652               | 0.083                |
> | 3    | 5.168                     | 0.153                     | **5.436**               | **0.158**                |
> | 5    | 4.608                     | 0.223                      | 4.838               | 0.187                |
>
>
> **Number of denoising steps and NFE configuration.** As shown in **Tab R1**, the number of denoising steps beyond 10 gives marginal gains as also observed in prior work on inference-time scaling in diffusion models [39]. Hence, we fixed the number of denoising steps to 10 to ensure fair and consistent evaluation across all methods. Note that once the number of denoising steps is fixed, the total particle count per step is automatically determined by dividing the total NFE budget by the number of steps.
>
> **Diffusion coefficient design and norm.** From the empirical experiments shown in **Tab R2**, $g(t)=3t^2$ yielded the most favorable balance between sample diversity and output fidelity. We observe that using too small norms failed to sufficiently diversify the particle set. Conversely, larger norms induced excessive noise, degrading image quality and alignment.
>
> **[W2: Clarification of contributions]**
>
> We appreciate the reviewer’s comment. We clarify that our contribution goes beyond simply adopting the VP-SDE interpolant, and instead centers on **analyzing and leveraging interpolant conversion to enable inference-time scaling in flow models**. To the best of our knowledge, this is the **first work to establish a connection between interpolant design and sample diversity**, and to provide both theoretical and empirical analysis supporting this link.
>
> Due to page limit, we provided a detailed analysis in **Appendix. C**, where we establish a formal connection between **interpolant choice and the variance of the proposal distribution**. In **Fig. 4 and 6**, we empirically showed that this variance directly affects the diversity of generated samples and thus the overall effectiveness of inference-time scaling. Additionally, in **our response to W2 of Reviewer kdqx**, we present an ablation study that isolates the impact of each component of interpolant conversion—timestep conversion and diffusion norm scaling on sample diversity. This study empirically confirms that both components synergistically increase sample diversity and sample quality.
>
> From our findings, we believe that taking the reverse approach—i.e., using a VP-SDE diffusion model but reverting to a Linear-SDE interpolant at inference—would **decrease sample diversity**, shrink the exploration space, and ultimately degrade the performance of inference-time scaling.
>
> ---
>
> **[W3: Non-uniform timestep schedule]**
>
> Thank you for highlighting this point—this direction indeed helps enrich our analysis. While the referenced works also employ non-uniform timestep scheduling, they require training for each model type.  This constraint prevented us from directly applying their methods in our setting. Also, it is important to note that their focus is primarily on ODE-based sampling, whereas our goal is to explore SDE-based inference to balance the trade-off between sample diversity and quality for efficient inference-time scaling.
>
> Instead, we include a quantitative comparison in **Tab. R3** using four different timestep schedulers including linear [32], ours, polynomial [32], and EDM [22]. Here, linear and polynomial are special case of setting $n=1$ and $n=2$ in $1-(1-t)^n$, respectively. Our results show that timestep schedules emphasizing finer integration in the early stages—specifically ours and polynomial—consistently yield better performance.
>
> **Tab. R3**: *Time scheduler comparison on Linear SDE*
>
> | Timestep Scheduler     | Diversity | Aesthetic Score | VQAScore | Inst. BLIP (Held-out) |
> |------------------------|-----------|------------------|---------------|------------------|
> | Linear ($n=1$) [32]               | 0.078   | 5.302          | 0.790       | 0.758          |
> | Ours     | **0.158**   | **5.436**          | **0.900**       | **0.813**          |
> | Polynomial ($n=2$) [32]     | 0.134   | 5.429          | 0.890       | 0.807          |
> | EDM  [22]     | 0.079   | 5.237          | 0.834       | 0.768         |
>
> ---
>
> **[W4: Inference-time cost]**
>
> Here, we provide a more comprehensive analysis in **Tab. R4**, where we evaluate runtime and performance across varying NFE budgets using NVIDIA L40S.
>
> **Tab. R4**: *Runtime and reward alignment under varying NFE budgets*
>
> | NFE  | Runtime (Sec) | VQAScore |
> |------|--------------|---------------|
> | 50   | 84.00        | 0.858       |
> | 100  | 140.11       | 0.882       |
> | 300  | 383.89       | 0.915       |
> | 500  | 635.01       | 0.925       |
> | 1000 | 1243.68      | 0.928       |
>
> As shown in **Tab. R4**, increasing the NFE budget leads to consistent performance improvements, demonstrating that RBF effectively scales with more compute. Importantly, this setup enables flexible trade-offs between runtime and reward alignment—users can adjust the NFE budget to meet their specific computational or fidelity requirements.
>
> ---
>
> **[W5: Component ablations and hyperparameter rationale]**
>
> We respectfully clarify that the main paper presents key ablations in **Fig. 4 and 6**, which demonstrate performance improvements from Linear-ODE to Linear-SDE, and further to VP-SDE across all particle-sampling-based methods. These figures also show gains from incorporating the rollover strategy (SVDD to RBF).
>
> To complement these results, we additionally present a more comprehensive ablation study in **Tab. R5**, where we vary the total NFE budget and isolate the impact of each core component—SDE conversion, interpolant conversion, and rollover strategy. For this experiment, we use the SVDD search strategy as the base, as RBF reduces to SVDD when the rollover mechanism is disabled.
>
> **Tab. R5**: *Ablation study across particle sampling variants*
>
> | Method            | NFE  | VQAScore | Inst.BLIP (Held-out) | Aesthetic |
> |-------------------|------|----------|------------------------|-----------|
> | BoN           | 100  | 0.846    | 0.796                  | 5.259     |
> |                   | 500  | 0.879    | 0.820                  | 5.162     |
> |                   | 1000 | 0.889    | 0.827                  | 5.205     |
> | SVDD (Linear-SDE) | 100  | 0.861    | 0.791                  | 5.045     |
> |                   | 500  | 0.893    | 0.813                  | 5.052     |
> |                   | 1000 | 0.892    | 0.820                  | 4.994     |
> | SVDD (VP-SDE)     | 100  | 0.874    | 0.809                  | 5.286     |
> |                   | 500  | 0.915    | 0.847                  | 5.249     |
> |                   | 1000 | 0.921    | 0.851                  | 5.281     |
> | RBF (VP-SDE)       | 100  | 0.882    | 0.821                  | 5.321     |
> |                   | 500  | 0.925    | 0.843                  | 5.237     |
> |                   | 1000 | 0.928    | 0.837                  | 5.259     |
>
> We conduct these experiments on the compositional text-to-image generation task, following the setup in the main paper. As shown in **Tab. R5**, we observe consistent improvements in reward alignment across all NFE regimes when moving from Linear-SDE to VP-SDE, and further with the addition of the rollover strategy (RBF). Notably, both VP-SDE SVDD and RBF outperform BoN across all metrics. We observe a similar trend in **Fig. 12** in Appendix for quantity-aware image generation, further supporting the generality of our findings. These results strongly validate the benefit of each proposed component and the modular design of our framework.
>
> Lastly, we kindly remind the reviewer that the rationale for key hyperparameter choices is presented in our previous responses to W1 and W3, including: the number of denoising steps (**Tab. R1**), diffusion coefficient (**Tab. R2**), and timestep scheduler (**Tab. R3**). We will incorporate these into the main paper to clearly justify our design choices.
>
> **[Q1: Non-uniform scheduler]**
>
> Yes, the observation aligns with our timestep scheduling described in Appendix D. The curved trajectories at the beginning would lead to greater diversity at the early stage, so we allocate more budget at the beginning accordingly.
>
> **[Q2: Applicability to other domains]**
>
> We thank the reviewer for this thoughtful suggestion. We fully agree that molecule and protein generation are promising domains for applying inference-time scaling. We will explore this direction and include additional results in the revision.

---

> > ### Comment · Reviewer_Hs6F · 2025-08-05
> >
> > Thanks to the authors for the response. Your answers resolve some of my concerns. I will revise the score accordingly.

---

> > > ### Author Response · Authors · 2025-08-06
> > >
> > > If there are any remaining concerns that may not have been fully resolved, could we respectfully ask the reviewer to share them? We would be eager to provide further clarification. We sincerely appreciate the reviewer for taking the time to review our response. We are grateful that some concerns have been addressed and that a score revision is under consideration.

---

> > > > ### Comment · Reviewer_Hs6F · 2025-08-07
> > > >
> > > > I have no further questions by now. I revised my evaluation based on your answers

---

### Official Review · Reviewer_kdqx · 2025-06-25

**Clarity:** 3
**Significance:** 2
**Originality:** 2
**Rating:** 4
**Confidence:** 3

**Summary:**

This paper considers inference time scaling for generating high quality images using flow models by leveraging the existing inference time scaling methods. The paper proposes three techniques to achieve the above purpose, which are (1) extending the deterministic ODE sampling to stochastic sampling, (2) transforming the stochastic sampling to VP-based stochastic sampling, and (3) Rollover Budget Forcing to dynamically allocate the computational resources over timesteps.

**Questions:**

See my comments above.

**Ethical Concerns:**

["NO or VERY MINOR ethics concerns only"]

**Final Justification:**

After checking the rebuttal responses from the authors, I decide to raise the score to bordeline accept. I hope the authors rewrite the paper to clarify their contributions as they promised.

**Limitations:**

The authors mentioned that their method introduces additional inference-time overhead. But it is not clear how much additional overhead is introduced by the method.

**Quality:**

2

**Strengths And Weaknesses:**

Strengths:
(1) The main contribution of the paper is to convert the deterministic ODE sampling of flow models to VP-based stochastic sampling to be able to use existing inference time scaling methods developed for diffusion models.
(2) Extensive experimental results confirms the effectiveness of the proposed techniques.

Weaknesses:
(1) As noted in the paper, the processes for convert the deterministic ODE sampling of flow models to stochastic sampling is already covered in the SiT [38]. Even though the SiT paper focus on improving sampling quality of flow models, the mathematical derivations for converting deterministic sampling of flow models to stochastic sampling are well explained. This makes Subsection 4.3 on Inference-Time SDE Conversion prior work instead of new contribution.

(2)  It is also known from literature [33] and [52] how to convert a stochastic sampling to a VP-based stochastic sampling.   I would suggest the authors to rewrite Subsection 4.3 on Inference-Time Interpolant Conversion in order to better explain why the conversion makes the search space larger.  In the paper, the authors simply state that "In Fig.3(c), we visualize the sample diversity under VP-SDE using FLUX which generates more diverse samples than Linear-SDE." My question is what the reason behind it. Can the authors provide some intuitive argument?

---

> ### Author Rebuttal · Authors · 2025-07-31
>
> **[W1: Clarification of our contributions]**
>
> We appreciate the reviewer for raising this important point. We would like to emphasize that our core contribution does not lie in re-deriving techniques such as SDE conversion, but rather in repurposing and integrating them through the lens of particle sampling to **enable efficient inference-time scaling for flow models**.
>
> To the best of our knowledge, this work is the first to show that inference-time particle sampling—traditionally exclusive to diffusion models—is both feasible and effective in flow models. This is a nontrivial insight: when using ODE-based sampling, all particles deterministically collapse into a single trajectory, rendering exploration effectively impossible. The introduction of an SDE injects variance into the proposal distribution, forming the basis for reward-guided sampling in flow models.
>
> Additionally, we show that SDE conversion alone is insufficient for achieving effective inference-time scaling. To address this, we introduce interpolant conversion as a key mechanism for increasing sample diversity—an insight presented for the first time in this work. In **Appendix C**, we provide a detailed analysis showing how interpolant design governs the variance of the proposal distribution, a factor central to exploration efficiency. Additionally, in **our response to W2**, we present an ablation study that empirically validates these findings. Together, these results establish a previously unexplored but critical connection between interpolant choice and particle sampling effectiveness in flow models.
>
> Our contributions span three key aspects:
> - We reinterpret SDE conversion as a tool for constructing an effective proposal distribution, **enabling efficient particle sampling in flow models**—a capability that was not previously possible.
>
> - To our knowledge, this is the **first work to view interpolant conversion from the perspective of sample diversity** and apply it for improving the efficiency of inference-time scaling. We present a detailed analysis that reveals how interpolant choice influences the sample diversity.
>
> - We introduce Rollover Budget Forcing (RBF), a novel strategy that **adaptively reallocates compute** across timesteps to maximize utilization of compute budget.
>
> We appreciate the reviewer’s feedback and will revise the main paper to clearly highlight our novel contributions and better distinguish them from prior work.
>
> **[W2: How interpolant conversion enhances sample diversity?]**
>
> We appreciate the reviewer’s insightful suggestion. Due to space constraints, our analysis of the relationship between interpolants and sample diveristy was provided in **Appendix. C**. We promise the reviewer that we will revise the main paper to incorporate both the analytical discussion from **Appendix C** and the supporting ablation study presented in this response.
>
> In **Appendix C**, we dissect interpolant conversion into its two core components—diffusion coefficient scaling and timestep conversion—and **analyze their impact on the variance of the proposal distribution**. Here, in this response, we present an ablation study that **explains how each component of interpolant conversion affects sample diversity**, validating our analysis in **Appendix C**.
>
> First, to deepen our understanding of why interpolant conversion is effective, we analyze the impact of **diffusion norm scaling** $g_{t_s}' = \frac{g_s}{c_s} \cdot \sqrt{\frac{\Delta s}{\Delta t_s}}$, which increases the stochasticity of the generative process since $c_s \approx 1$ and $\sqrt{\frac{\Delta s}{\Delta t_s}} \gg 1$. While this scaling increases the variance of the proposal distribution and enhances sample diversity, it can also inject excessive noise relative to the current timestep. This may result in detached samples from well-supported regions of the sample manifold—especially for sharp distributions (e.g., small timesteps)—ultimately **harming sampling performance**.
>
> In contrast, **timestep conversion** shifts sampling toward earlier timesteps, ($s < t_s$) where the noise level is higher and the distribution is more diffuse. In principle, this could promote diversity; however, it also reduces the integration step size $\Delta t_s$ (see Fig 8, $\Delta s$ corresponds to smaller $\Delta t_s$ at early steps), thereby lowering the actual variance of the proposal distribution, given by $g_{t_s}^2 \Delta t_s$. This diminished variance counteracts the intended benefit of enhanced exploration, leading to only minor improvements in reward alignment and diversity.
>
> These findings suggest that diffusion norm scaling and timestep conversion are not independently sufficient, but rather act as complementary mechanisms. Interpolant conversion—as implemented in VP-SDE—combines both:
> - It introduces stronger stochasticity via norm scaling,
> - While stabilizing the sampling trajectory through an annealed timestep schedule.
>
> To further validate this, we present an ablation study by isolating and evaluating each component separately.
>
> **[Ablation study]**
>
> **Tab. R1**: *Ablation on interpolant conversion strategies*
> | Method                      | Diversity ↑ | Aesthetic ↑ | VQAScore ↑ | Inst. Blip (Held-out) ↑ |
> |----------------------------|-------------|--------------|-----------------|--------------------|
> | Linear-SDE                 | 0.158       | 5.436        | 0.900           | 0.813              |
> | + Adapt. Diffusion         | 0.429       | 3.986        | 0.702           | 0.571              |
> | + Adapt. Timestep          | 0.270       | 5.487        | 0.908           | 0.813              |
> | VP-SDE (Ours)              | **0.509**   | **5.830**    | **0.925**       | **0.843**          |
>
> **Tab. R1** isolates the effects of each component (stochasticity and interpolant conversion) and evaluates their impact on:
> - Sample diversity (LPIPS-MPD [23]),
> - Sample quality (Aesthetic Score),
> - Reward alignment (using the compositional text-to-image task in the main experiments).
>
> All metrics are computed using 1,000 generated images. To the best of our knowledge, this is the first work to explicitly analyze how these components interact and influence the sample diversity and exploration capability of flow-based generative models.
>
> In **Tab. R1**, we see that applying diffusion norm scaling alone (“+ Adapt. Diffusion”) results in a noticeable increase in sample diversity, but also leads to a significant drop in both reward alignment and sample quality—likely due to the injection of excessive noise. In contrast, timestep conversion (“+ Adapt. Timestep”) avoids such degradation but yields only modest improvements in diversity. The VP-SDE variant demonstrates that the coordinated use of both components can expand the exploration space without deviating from the underlying data manifold, achieving both high diversity and high fidelity.
>
> In this work, we primarily focused on the interpolant side and analyzed its connection to timestep conversion and diffusion norm scaling. As noted in **Appendix C**, exploring the reverse direction—designing new interpolants based on desired variance properties of the proposal distribution—would also be an interesting future direction, and we plan to take concrete steps toward this in follow-up work.
>
> **[L1: Inference-time overhead]**
>
> Thank you for pointing this out. While we report the runtime overhead of RBF under a 500-NFE budget in **Appendix F.3**, we provide a more comprehensive analysis in **Tab. R2**, where we evaluate runtime using NVIDIA L40S GPU.
>
> **Tab. R2**: *Runtime and known reward under different NFE budgets*
>
> | NFE Budget | Runtime (Sec) | VQAScore ↑ |
> |------------|----------------|----------------|
> | 50         | 84.00          | 0.858        |
> | 100        | 140.11         | 0.882        |
> | 300        | 383.89         | 0.915        |
> | 500        | 635.01         | 0.925        |
> | 1000       | 1243.68        | 0.928        |
>
> As shown in **Tab. R2**, increasing the NFE budget leads to consistent improvements in reward alignment, demonstrating that RBF effectively scales with more compute. Importantly, this setup enables a **flexible trade-off between runtime and performance**—users can adjust the NFE budget according to their computational constraints or desired fidelity level.

---

> > ### Comment · Reviewer_kdqx · 2025-08-05
> >
> > After checking the rebuttal responses from the authors, I decide to raise the score to bordeline accept. I hope the authors rewrite the paper to clarify their contributions as they promised.

---

> > > ### Author Response · Authors · 2025-08-06
> > >
> > > We sincerely thank the reviewer for taking the time to review our response and raising the score. As promised, we will revise the paper to more clearly show our contributions. We truly appreciate your constructive feedback throughout the process.

---

### Official Review · Reviewer_P9ao · 2025-07-02

**Clarity:** 3
**Significance:** 3
**Originality:** 2
**Rating:** 4
**Confidence:** 4

**Summary:**

The paper introduces VP‑SDE, an inference‑time scaling framework for pretrained flow models that brings the benefits of particle sampling—previously exploited in diffusion models—to the deterministic flow setting. First, it casts flow generation as an SDE, enabling Monte Carlo sampling of multiple trajectories. Second, it enlarges the candidate output space via a Variance‑Preserving (VP) interpolant conversion. Third, it allocates computational budget adaptively across timesteps through Rollover Budget Forcing (RBF), dynamically adjusting the number of function evaluations (NFEs). Empirical results on Compositional Text‑to‑Image Generation and Quantity‑Aware Image Generation demonstrate that VP‑SDE with RBF outperforms prior inference‑time scaling methods for flow and diffusion models.

**Questions:**

1. Can you provide a detailed time‑ and space‑complexity analysis of each sampling strategy (e.g. vanilla flow, SDE‑based, VP interpolant, RBF) under a fixed compute budget? In particular, is Rollover Budget Forcing (RBF) truly optimal in all regimes, or only once you scale up the model?

2. The phrase “much clearer posterior mean” on line 241 is vague—what metric or visualization supports this description?

3. Could you supply a comprehensive list of all critical hyperparameters (e.g. number of particles, RBF schedule parameters, step size for SDE integration) and the rationale behind their chosen values?

4. How sensitive are your results to the number of particles and the specific RBF allocation schedule?

5. Have you tested VP‑SDE and RBF on other flow model variants (e.g. neural ODE flows, continuous normalizing flows) or non‑image tasks (e.g. audio, tabular data)?

**Ethical Concerns:**

["NO or VERY MINOR ethics concerns only"]

**Final Justification:**

The authors clarified their contributions and conducted a comprehensive ablation study to address my concerns about weaknesses; they also answered my questions regarding confusion and paper details. I think this is a good work on exploring the integration of variance into flow models, only with some ambiguity and lack of evidence for specific claims, so I recommend Borderline accept.

**Limitations:**

yes

**Quality:**

3

**Strengths And Weaknesses:**

## Strengths

- The paper pioneers the integration of particle sampling into flow matching models by reformulating the deterministic probability path evolution as a stochastic process, achieving a more effective exploration–exploitation trade‑off.

-  The manuscript is well organized and reads smoothly; theoretical derivations are detailed and easy to follow.

-  The supplementary material is thorough and supports the main text with additional proofs and experimental details.

## Weaknesses

-  Prior works (e.g., SiT) have already applied very similar techniques to improve the sampling process; this paper adopts those ideas almost unchanged, resulting in limited novelty.

-  While the authors claim that stochastic, variance‑preserving sampling strategies help discover high‑reward, low‑density regions, they provide neither rigorous theoretical justification nor dedicated ablation studies to substantiate this claim, leaving the exploration benefit insufficiently supported.

---

> ### Author Rebuttal · Authors · 2025-07-30
>
> **[W1: Clarification of our contributions]**
>
> We would like to emphasize that our core contribution does not lie in re-deriving techniques such as SDE conversion, but rather in repurposing and integrating them through the lens of particle sampling to **enable efficient inference-time scaling for flow models**.
>
> To the best of our knowledge, this work is the first to show that **inference-time particle sampling—traditionally exclusive to diffusion models—is both feasible and effective in flow models**. This is a nontrivial insight: when using ODE-based sampling, all particles deterministically collapse into a single trajectory, rendering exploration effectively impossible. The introduction of an SDE injects variance into the proposal distribution, forming the basis for reward-guided sampling in flow models.
>
> Additionally, we observe that SDE conversion alone is insufficient for achieving effective inference-time scaling. To address this, we establish— **for the first time— a connection between interpolant design and sample diversity**. Our work further provides a **principled analysis of how interpolant design governs the variance of the proposal distribution** (see **Appendix C**). This analysis is supported by an ablation study included in the **response to W2 of Reviewer kdqx**, where the impact of each component of interpolant conversion on sample diversity is analyzed.
>
> Our contributions span three key aspects:
> - We reinterpret SDE conversion as a tool for constructing an effective proposal distribution in particle sampling, **enabling efficient inference-time scaling in flow models**.
>
> - To our knowledge, this is the **first work to view interpolant conversion from the perspective of sample diversity** and apply it for improving the efficiency of inference-time scaling. We present a detailed analysis that reveals how interpolant conversion influences the sample diversity.
>
> - Additionally, we introduce Rollover Budget Forcing (RBF), a novel particle sampling strategy that **adaptively reallocates compute** across timesteps to maximize utilization of compute budget.
>
> ---
>
> **[W2: Ablation study of SDE and interpolant conversion in sample diversity and reward alignment]**
>
> **Tab. R1**: *Ablation study of SDE and interpolant conversion*
>
> | Method                      | Diversity ↑ | Aesthetic ↑ | VQAScore ↑ | Inst. BLIP (Held-out) ↑ |
> |----------------------------|-------------|--------------|-----------------|--------------------|
> | Linear-ODE                 | –           | 5.624        | 0.788           | 0.789              |
> | Linear-SDE                 | 0.158       | 5.436        | 0.900           | 0.813              |
> | + Adapt. Diffusion         | 0.429       | 3.986        | 0.702           | 0.571              |
> | + Adapt. Timestep          | 0.270       | 5.487        | 0.908           | 0.813              |
> | VP-SDE (Ours)              | **0.509**   | **5.830**    | **0.925**       | **0.843**          |
>
> We thank the reviewer for raising this important point. We present empirical results on how increased diversity can enhance reward alignment in **Fig. 4 and 6**. To further support these findings, we include additional quantitative results in **Tab. R1**, which substantiate how both SDE and interpolant conversion contribute to improved sample diversity and reward alignment under a controlled setup. We measure diversity using LPIPS-MPD [23], sample quality using the Aesthetic Score on 1,000 generated images.
>
> We first highlight that Linear-SDE significantly increases sample diversity over Linear-ODE, thereby expanding the exploration space and improve reward alignment. Building on this, we observe that VP-SDE further enhances sample diversity and expands the exploration space, leading to improved reward alignment. This improvement stems directly from interpolant conversion, which integrates two key mechanisms analyzed in **Appendix C**: diffusion norm scaling and timestep conversion. Due to space constraints, we refer the reviewer to our **response to W2 of Reviewer kdqx** for a more detailed analysis of how each component contributes to sample diversity.
>
> ---
>
> **[Q1: Time complexity and RBF scaling]**
>
> We clarify that the RBF scaling in quantity-aware image generation is presented in **Fig. 12**. Here, we include time complexity analysis and RBF scaling in compositional text-to-image generation.
>
> **Tab. R2**: *Inference cost*
>
> | Method         | Inference Cost                                   |
> |----------------|--------------------------------------------------|
> | Base           | $S \cdot c_d$                                    |
> | BoN            | $N \cdot c_d + \frac{N}{S} \cdot c_v$            |
> | SMC            | $N \cdot c_d + N \cdot c_v$                      |
> | RBF, SVDD     | $N \cdot c_d + N \cdot c_v$                      |
>
> - $S$: Denoising steps
> - $N$: NFE budget
> - $c_d$, $c_v$: Cost of denoising and verification (reward)
>
> As shown in **Tab. R2**, since all methods share the same NFE budget, the cost for denoising is set to $N * c_d$.
> The verification cost varies across methods. Notably, both SVDD and RBF consistently outperform BoN across all NFE budget regimes (**Fig. 12 and Tab. R3**), despite incurring only marginal overhead in verification cost. In terms of space complexity, all methods have similar memory usage, with the flow model accounting the most (32GB). We appreciate the comment and will revise the paper to include the analysis of time complexity.
>
> **Tab. R3**: *Scaling comparison*
>
> | Method           | NFE | VQAScore | Inst.Blip (Held-out) | Aesthetic |
> |------------------|-----|-----------|------------------------|------------------|
> | **BoN**     | 100 | 0.846     | 0.796                  | 5.259            |
> |                  | 500 | 0.879     | 0.820                  | 5.162            |
> |                  | 1000| 0.889     | 0.827                  | 5.205            |
> | **SVDD (Linear-SDE)** | 100 | 0.861     | 0.791                  | 0.809            |
> |                       | 500 | 0.893     | 0.813                  | 5.052            |
> |                       | 1000| 0.892     | 0.820                  | 0.851            |
> | **SVDD (VP-SDE)**     | 100 | 0.874     | 0.809                  | 5.286            |
> |                       | 500 | 0.915     | 0.847                  | 5.249            |
> |                       | 1000| 0.921     | 0.851                  | 5.281            |
> | **RBF (VP-SDE)**     | 100 | 0.882     | 0.821                  | 5.321            |
> |                  | 500 | 0.925     | 0.843                  | 5.237            |
> |                  | 1000| 0.928     | 0.837                  | 5.259            |
>
> The results in **Fig. 12 and Tab. R3** show that RBF maintains strong performance across all NFE regimes. Additionally, we note that **the total number of particles per step is automatically determined for all baselines as NFE budget divided by denoising steps**, as described in **Appendix. E**. We further emphasize that RBF introduces no additional hyperparameters. The initial NFE quota is uniformly allocated across all steps from the total budget, and RBF dynamically reallocates any unused quota based on the discovery of higher-reward samples.
>
> ---
>
> **[Q2: Ambiguity of “much clearer posterior mean”]**
>
> We agree that the phrase could be more precise. Flow models trained with rectification exhibit more **disentangled trajectories**, allowing their velocity predictions to be computed with minimal entanglement across samples. This reduces averaging artifacts and blurry posterior means often seen in diffusion models with overlapping trajectories.
>
> As a result, the posterior mean $\mathbb{E}[x_0 \mid x_t]$ (Tweedie’s formula) is more accurate estimate of the final sample, improving reward prediction during inference. We will clarify this point and include supporting visuals in the final version.
>
> ---
>
> **[Q3: Hyperparameter choice and rationale]**
>
> We clarify our choice of hyperparameters. We set denoising steps to 10 for all experiments and diffusion coefficient $g(t) = 3t^2$. Below we provide rationale of this choice.
>
> **Tab. R4**: *Number of denoising step*
> | Steps | Aesthetic | Diversity |
> |-------|------------------|-----------|
> | 10    | **5.635**            | **0.084**     |
> | 20    | 5.680            | 0.103     |
>
> **Tab. R5**: *Diffusion coefficient*
> | Norm | Aesthetic $g(t)=t$ | Diversity $g(t)=t$ | Aesthetic $g(t)=t^2$ | Diversity $g(t)=t^2$ |
> |------|---------------------------|----------------------------|---------------------|----------------------|
> | 1    | 5.635                     | 0.084                      | 5.652               | 0.083                |
> | 3    | 5.168                     | 0.153                      | **5.436**               | **0.158**                |
> | 5    | 4.608                     | 0.223                      | 4.838               | 0.187                |
>
> As shown in **Tab R4**, the number of denoising steps beyond 10 gives marginal gains. Hence, we fixed the number of denoising steps to 10 to ensure fair and consistent evaluation across all methods. Note that once the number of denoising steps is fixed, the total particle count per step is automatically determined by dividing the total NFE budget by the number of steps.
>
> **Tab. R5** shows sample diversity and sample quality with different diffusion coefficient. We found that using $ g(t) = 3t^2 $ consistently yielded the best balance between sample diversity and output fidelity across tasks.
>
> ---
>
> **[Q4: Particle size and RBF schedule]**
>
> We clarify that varying the particle size is equivalent to scaling the NFE budget. Please see our **response to Q1** for details.
>
> ---
>
> **[Q5: Other flow models and domains]**
>
> We agree that the proposed method is broadly applicable to other modalities—such as molecular generation, audio, or tabular data—and we are planning to extend our framework to these domains.

---

> > ### Comment · Reviewer_P9ao · 2025-08-06
> >
> > Many thanks to the authors for the detailed explanations and fruitful experiments. I hope the authors apply the revisions mentioned above in the updated version. I remain positive about this work. One interesting point in W2 is that performance seriously drops when adding Adaptive Diffusion to Linear-SDE, so I'm wondering if simply integrating Adaptive Timestep into Linear-ODE could result in better performance than VP-SDE?

---

> > > ### Author Response · Authors · 2025-08-06
> > >
> > > We sincerely thank the reviewer for the thoughtful comments and continued positive assessment of our work. We would like to clarify that Linear-ODE follows a **deterministic** trajectory determined by the initial Gaussian noise, yielding identical outputs across runs—regardless of the timestep scheduler—aside from minor discretization errors. As such, applying an Adaptive Timestep schedule does not introduce any stochasticity; the variance of the proposal distribution remains zero, and thus the sample diversity does not change. Consequently, this modification does not lead to meaningful performance changes to the original Linear-ODE setup, which underperforms compared to stochastic counterparts such as VP-SDE.

---

### Official Review · Reviewer_tyvx · 2025-07-20

**Clarity:** 4
**Significance:** 3
**Originality:** 2
**Rating:** 5
**Confidence:** 3

**Summary:**

This paper adapts particle sampling to flow-matching models. Particle sampling consists of generating several candidates along the denoising process and picking the best according to a defined reward function. Flow-matching models are deterministic, and thus cannot generate several candidates to enable particle sampling. To address this, the paper proposes two changes to the sampling process of flow-matching models: it derives the stochastic path corresponding to the flow-matching ODE (this introduces the stochasticity needed to generate several candidates) and converts the uniform interpolant to the VP interpolant commonly used in diffusion models (this increases the variance of the generated candidates). On top of these changes, the authors propose a greedy budget-allocation strategy that greedily distributes the number of particles generated at each denoising time-step. These 3 changes are evaluated and perform well on compositional text-to-image generation and quantity-aware generation.

**Questions:**

1. The greedy budget re-allocation scheme is interesting. However, from Figure 9, it seems that it is "rare" for the `time of occurrence of higher reward` to exceed the uniform budget. If the authors were to remove the budget reallocation strategy but simply cut the uniform budget early (i.e removing line 7 in alg.2) does RBF simply become CODE? Please correct my understanding if not. Moreover, it appears that RBF is harder to parallelize compared with other search schemes. Could the authors discuss this some more?

**Ethical Concerns:**

["NO or VERY MINOR ethics concerns only"]

**Limitations:**

A limitation the authors should discuss is the aesthetics score, which appears to be harmed by most of the inference time modifications.

**Paper Formatting Concerns:**

No concerns.

**Quality:**

4

**Strengths And Weaknesses:**

## Strengths
- The paper is very well written and builds up its ideas in a clear and understandable manner. Synthetic examples provide good illustrations.
- The proposed ideas, namely VP-SDE sampling of flow models performs well, compared to the baseline, non-stochastic or linear variants, in some cases.

## Weaknesses
1. The comparison with diffusion models needs more discussion: the authors effectively propose a method diffusion-ify a pre-trained flow model. The comparisons should be between a particle sampling method applied to diffusion models vs the authors' VP-SDE flow models. The authors provide a table in the appendix comparing base diffusion and base flow models but the more interesting comparison is one where both are using particle-sampling. Does the NFE advantage of flow models persist in that setting as well? Is the paper's message that if one wants to do inference-time scaling, then convert the flow-model to a diffusion model at inference time? Moreover, some care is necessary when comparing more recent, larger flow models with older diffusion models.
2. The two main ingredients of the paper which are:  1)changing between deterministic ODEs and stochastic paths and 2) interpolant conversion of a pre-trained model are known results. The combination and the end-goals are different, which is why I still recommend acceptance. However, the method is not adapting particle sampling to flow-models, but rather making flow-models as diffusion model like as possible to apply existing inference time-scaling methods.
---
minor point: The Laplacian symbol $\Delta$ might be better to avoid confusion with the Hessian symbol in the proof of Proposition 1.

---

> ### Author Rebuttal · Authors · 2025-07-30
>
> **[W1: Comparison with Diffusion Models under Particle Sampling]**
>
> We sincerely thank the reviewer for this insightful point regarding comparisons with diffusion models.
>
> While an ideal comparison would involve models trained under identical architecture and dataset conditions, such training is resource-intensive. Instead, we chose the most competitive available models: **FLUX** (flow model, Aug. 2024) and **SANA** (diffusion model, Jan. 2025), both trained on high-quality datasets. We acknowledge that this may appear to be an unfair comparison, which is why we omitted it from the main paper. However, we include it here as a reference point to illustrate the distinct advantages that flow models can offer over diffusion models.
>
> To address the reviewer’s concern, we apply **particle sampling to both models**, where ×2 and ×3 denote increased compute budgets for the denoising steps. As shown in **Tab. R1**, flow model consistently outperforms SANA across all settings:
>
> **Tab. R1**: *Comparison of particle sampling between SANA (diffusion) and FLUX (flow)*
> | NFE         | SANA (1×) | SANA (2×) | SANA (3×) | FLUX (Ours) |
> |-------------|-----------|-----------|-----------|----------------------|
> | VQAScore ↑  | 0.682     | 0.695     | 0.706     | **0.925**            |
> | InstBLIP ↑  | 0.749       | 0.737      | 0.745    | **0.843**            |
> | Aesthetic ↑ | 5.288     | 5.338     | 5.354     | **5.237**            |
>
> This confirms that our approach enables effective inference-time scaling in flow models, not just mimicking diffusion, but actually leveraging the structural advantages of flow models for superior performance. Notably, reflow training in flow models reduces trajectory entanglement, ensuring that each sampling path remains well-separated. When trajectories remain disentangled, the velocity at each intermediate state is predicted based solely on the local sample path, rather than being implicitly averaged across nearby trajectories. This leads to more accurate predictions that point toward the target data sample, rather than toward a smoothed or interpolated region. As a result, the posterior mean estimates are more accurate and less prone to blurring, improving the reliability of reward evaluation in particle sampling. While we briefly mention this in L239–L242, we will revise the final version to ensure that this distinction is better presented.
>
> In short, our work does not simply convert flow models into diffusion models, but instead expands their inference-time capabilities while preserving their core strengths—offering distinct advantages for efficient scaling.
>
> ---
>
> **[W2: Clarification on "making flow-models as diffusion model like as possible"]**
>
> Thank you for recognizing that the combination and end goals of our approach differ from those of previous works. In response to your comment, we would like to highlight the following:
>
> 1. Applying particle sampling to a flow model critically requires the ODE-to-SDE conversion, as stochastic sampling at intermediate steps is essential for particle sampling. More importantly, we have shown in this work that **inference-time particle sampling—traditionally exclusive to diffusion models—is both feasible and effective in flow models**.
>
> 2. Our use of interpolant conversion from Linear-SDE to VP-SDE was not merely intended to “diffusion-ify” a flow model. Rather, it was the result of empirically identifying an interpolant that provides sufficient diversity during intermediate sampling without sacrificing output quality. To our knowledge, we are also the **first to demonstrate that the diversity of samples can be controlled by navigating the interpolant space**. We emphasize that our work offers a **principled analysis of how interpolant design governs the variance of the proposal distribution** (**Appendix C**), a connection which was not explored before. Additionally, we present an ablation study in the **response to W2 of Reviewer kdqx**, where the impact of each component of interpolant conversion on sample diversity is systematically evaluated.
>
> ---
>
> **[Q1: RBF reduces to SVDD without rollover strategy]**
>
> If we remove the budget reallocation mechanism (e.g., **Line 7 in Alg. 2**), RBF reduces to SVDD. CoDe runs verification (reward) only at periodic timesteps, whereas SVDD performs candidate selection at every step.
>
> We would like to clarify two key points regarding the behavior of RBF.
>
> First, **Fig. 9** illustrates the standard deviation ($\sigma$) of the required NFEs to discover a higher-reward sample. Importantly, as more prompts are evaluated, this leads to an increasing number of cases where the optimal timestep requires more NFEs than the uniform allocation. In such cases, RBF’s dynamic reallocation becomes critical for efficiently identifying higher-reward particles.
>
> Second, while RBF may seem less parallelizable as it involves a sequential component due to its greedy budget reallocation, in practice, RBF can be parallelized efficiently by increasing the batch size (the number of initial samples). In contrast, methods such as SoP and SMC are inherently more difficult to parallelize, as they require correction or resampling steps that involve cross-sample communication, necessitating complex engineering for synchronization.
>
> ---
>
> **[L1: Aesthetic Score Drops]**
>
> We appreciate the reviewer’s thoughtful observation. Indeed, reward over-optimization is a well-known issue in reward alignment tasks, where aggressively optimizing toward a given reward can lead to a significant drop in other metrics (e.g., aesthetic quality).
>
> In our experiments, while we observe a slight drop in aesthetic score, we emphasize that the **degradation is minor and does not result in noticeable artifacts or perceptual quality loss**. As illustrated in Fig. 5 and 7, the generated images remain sharp and visually compelling—even as alignment performance improves significantly (x6 accuracy improvement in quantity-aware image generation)—demonstrating the robustness of our approach.
>
> Additionally, we clarify that this trade-off is a **controllable design choice**: users can moderate the level of reward optimization by reducing the NFE budget to suit downstream needs. Overall, our results show that inference-time scaling enables high-impact improvements in alignment while preserving the expressive priors of flow models.

---

> > ### Author Response · Authors · 2025-08-06
> >
> > We sincerely thank the reviewer for taking the time to review our work. As the discussion period approaches its conclusion, we would like to respectfully ask if there are any remaining questions or suggestions. We would appreciate any additional input and remain fully committed to addressing any feedback.

---

### Note · Authors · 2025-08-12

Dear AC and Reviewers,

We sincerely thank all the reviewers for their insightful comments and for positively recognizing the contributions of our work (Reviewers tyvx, P9ao, and kdqx). We are pleased to have addressed all concerns raised by Reviewers P9ao and kdqx. Although we did not have the opportunity to engage in further discussion with Reviewer tyvx, we have provided additional experiments and clarifications to address the reviewer's comments. We hope the reviewer will have the chance to review our responses during the remainder of the discussion period.

Below, we summarize our responses addressing the reviewers’ concerns:
1. **Contributions of our work**: We repurpose and synergistically integrate SDE and interpolant conversion through the lens of particle sampling to enable efficient inference-time scaling for flow models. To our knowledge, this work is the first to show that inference-time particle sampling—previously exclusive to diffusion models—is both feasible and effective in flow models.
2. **Interpolant conversion and sample diversity**: In Appendix C, we establish a connection between interpolant conversion and sample diversity through two key components: diffusion norm scaling and timestep conversion. Additionally, in **our response to Reviewer kdqx**, we present ablation studies that quantify the individual and combined effects of these components on sample diversity. To our knowledge, this is the first work to explicitly reveal and systematically analyze this connection.
3. **Additional experiments**: Extending the results of Appendix F.3, we validated the superior scaling behavior of our method compared to Best-of-N, SVDD (Linear-SDE), and SVDD (VP-SDE) in **our response to Reviewer P9ao**. We also provided quantitative results on the compute–reward trade-off in **our responses to Reviewers kdqx and Hs6F**, and presented comprehensive experiments validating our choice of hyperparameters—diffusion coefficient, number of denoising steps, and timestep scheduler—in **our responses to Reviewers P9ao and Hs6F**.

We believe that our responses and dedicated experiments help clarify the novelty and effectiveness of our work, and also address the concerns raised by the reviewers.

---

### Decision · Program_Chairs · 2025-09-17

**Decision:**

Accept (poster)

**Comment:**

(a) The paper proposes VP-SDE, a framework to adapt particle sampling to pre-trained flow models. This allows inference-time scaling previously limited to diffusion models. Contributions include: i) converting deterministic flow ODEs to SDEs, ii) replacing uniform with variance-preserving (VP) interpolants for diversity, and iii) Rollover Budget Forcing (RBF) for adaptive budget allocation. Experiments on compositional and text-to-image tasks show consistent improvements.

(b) It has a clear and well-structured presentation, it extends inference-time scaling to flow models, its performance shows improvements.

(c) Reviewers found the paper to have a limited novelty: they point out that SDE conversion and VP interpolants are established ideas; the contribution seems to lie in combining them for flows. There are baseline gaps: Missing direct comparison with diffusion models using particle sampling. In terms of efficiency, it is not clear how scalable the method is (the added inference-time cost should be better quantified). Some hyperparameter choices lack justification.

(d) Nonetheless, the paper is found overall technically sound, clearly presented, and practically useful. The introduction of particle sampling to flow models is demonstrated satisfactorily. While the novelty is incremental and the evaluation incomplete, the contribution is meaningful for the generative modeling community.

e) Reviewers initially questioned originality, fairness of baselines, and clarity. The rebuttal improved the presentation, added ablations, and resolved many concerns, leading to several score increases. The remaining issues include novelty framing and missing efficiency analysis.

Thus, overall the paper has been found to be suitable for NeurIPS, as its practical contributions outweigh its incremental novelty.